# Deep Unlearning: Fast and Efficient Gradient-Free Class Forgetting

**Sangamesh Kodge**                                  *skodge@purdue.edu*
*Elmore Family School of Electrical and Compute Engineering,*
*Purdue University*

**Gobinda Saha**                                     *gsaha@purdue.edu*
*Elmore Family School of Electrical and Compute Engineering,*
*Purdue University*

**Kaushik Roy**                                      *kaushik@purdue.edu*
*Elmore Family School of Electrical and Compute Engineering,*
*Purdue University*

**Reviewed on OpenReview:** *https://openreview.net/forum?id=BmI5p6wBi0*

## Abstract

Machine *unlearning* is a prominent and challenging field, driven by regulatory demands for user data deletion and heightened privacy awareness. Existing approaches involve retraining model or multiple finetuning steps for each deletion request, often constrained by computational limits and restricted data access. In this work, we introduce a novel class unlearning algorithm designed to strategically eliminate specific classes from the learned model. Our algorithm first estimates the Retain and the Forget Spaces using Singular Value Decomposition on the layerwise activations for a small subset of samples from the retain and unlearn classes, respectively. We then compute the shared information between these spaces and remove it from the forget space to isolate class-discriminatory feature space. Finally, we obtain the unlearned model by updating the weights to suppress the class discriminatory features from the activation spaces. We demonstrate our algorithm's efficacy on ImageNet using a Vision Transformer with only $\sim 1.5\%$ drop in retain accuracy compared to the original model while maintaining under $1\%$ accuracy on the unlearned class samples. Furthermore, our algorithm exhibits competitive unlearning performance and resilience against Membership Inference Attacks (MIA). Compared to baselines, it achieves an average accuracy improvement of $1.38\%$ on the ImageNet dataset while requiring up to $10\times$ fewer samples for unlearning. Additionally, under stronger MIA attacks on the CIFAR-100 dataset using a ResNet18 architecture, our approach outperforms the best baseline by $1.8\%$. Our code available on github.[1]

## 1 Introduction

Machine learning has automated numerous applications in various domains, including image processing, language processing, and many others, often surpassing human performance. Nevertheless, the inherent strength of these algorithms, which lies in their extensive reliance on training data, paradoxically presents potential limitations. The literature has shed light on how these models behave as highly efficient data compressors (Tishby & Zaslavsky, 2015; Schelter, 2020), often exhibiting tendencies toward the memorization of full or partial training samples (Arpit, 2017; Bai et al., 2021). Such characteristics of these algorithms raise significant concerns about the privacy and safety of the general population. This is particularly concerning

---

[1]Our code is available at `https://github.com/sangamesh-kodge/class_forgetting`.

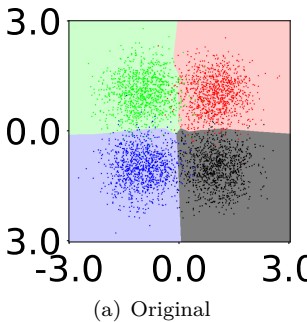
(a) Original

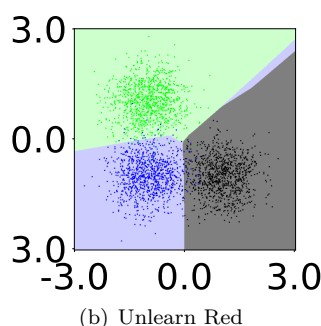
(b) Unlearn Red

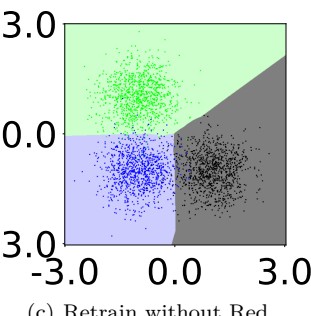
(c) Retrain without Red

**Figure 1:** Illustration of the unlearning algorithm on a simple 4 class classification problem. Figure shows the decision boundary for (a) original model, (b) our unlearned model redistributing the space to nearby classes and (c) retrained model without red class.

given that the vast training data, typically collected through various means like web scraping, crowdsourcing, and more, is not immune to personal and sensitive information. The growing awareness of these privacy concerns and the increasing need for safe deployment of these models have ignited discussions within the community and, ultimately, led to some regulations on data privacy, such as Voigt & Von dem Bussche (2017); Goldman (2020). These regulations allow the use of the data with the mandate to delete personal information about a user if they choose to opt out from sharing their data. The mere deletion of data from archives is not sufficient due to the memorization behavior of these models. This necessitates machine unlearning algorithms that can remove the influence of requested data or unlearn those samples from the model. A naive approach, involving the retraining of models from scratch, guarantees the absence of information from sensitive samples but is often impractical, especially when dealing with compute-intensive state-of-the-art (SoTA) models like ViT(Dosovitskiy et al., 2020). Further, efficient unlearning poses considerable challenges, as the model parameters do not exhibit a straightforward connection to the training data (Shwartz-Ziv & Tishby, 2017). In many practical scenarios, full dataset access is often restricted since companies typically only utilize the trained model in their applications. Additionally, data access is usually chargeable for each instance, making repeated access for unlearning purposes prohibitively expensive. Public datasets may also become unavailable for reuse in the future, and maintaining a copy of all training data is often costly and impractical. Furthermore, in the context of removing confusing or manipulated data from the model Goel et al. (2024); Schoepf et al. (2024), where the goal is to improve model generalization when the training dataset may contain misleading samples, obtaining the entire confusing/manipulated dataset is challenging and often compute-intensive. This limited access to data introduces additional challenges for the unlearning process. Consequently, several unlearning works explore sample-efficient approaches Golatkar et al. (2020a); Nguyen et al. (2020); Chundawat et al. (2023); Jeon et al. (2024).

Our work focuses on challenging scenarios of class unlearning and multi-class unlearning (task unlearning) (Golatkar et al., 2020a; 2021). For a class unlearning setup, the primary goal of the unlearning algorithm is to eliminate information associated with a target class from a pretrained model. This target class is referred to as the forget class, while the other classes are called the retain classes. The unlearning algorithm should produce parameters that are functionally indistinguishable from those of a model retrained without the target class. The key challenges in such unlearning are three folds (i) pinpointing class-specific information within the model parameters, (ii) updating the weights in a way that effectively removes target class information without compromising the model's usability on other classes and (iii) demonstrating scalability on large scale dataset with well trained models. In this work, we ask the question *"Can we unlearn class (or multiple classes) from a well-trained model given access to **few samples** from the training data on a large dataset?"*. Having a few samples is particularly important if the unlearning algorithms have to efficiently scale to large datasets having many classes to ensure fast and resource-efficient unlearning.

We draw insights from work by Saha et al. (2021) in the domain of continual learning, where the authors use the Singular Value Decomposition (SVD) technique to estimate the gradient space essential for the previous task and restrict future updates in this space to maintain good performance on previous tasks. This work

demonstrates a few samples (about 125 samples per task) are sufficient to obtain a good representation of the gradient space. Our work proposes to strategically eliminate the class discriminatory information by updating the model weights to maximally suppress the activations of the samples belonging to unlearn (forget) class. We start by estimating the Retain and the Forget Spaces using Singular Value Decomposition on the layerwise activations for the samples from the retain and unlearn classes, respectively. These spaces represent the feature or activation basis vectors for samples from the retain and forget classes. We compute the shared information between these spaces and remove it from the forget space to isolate class-discriminatory feature space. Finally, we obtain the unlearned model by updating the weights to suppress the class discriminatory features from the activation spaces. In a class-unlearning scenario, our algorithm competes with or outperforms several algorithms studied in this work, despite having access to very few samples from the training dataset (less than 4% for all our experiments). As our algorithm relies on very few samples from the training dataset it efficiently scales to large datasets like ImageNet (Deng et al., 2009), where we demonstrate the results using only $\sim 1500$ samples (0.0012% of the training dataset).

Contrary to our method, traditional unlearning methods (Tarun et al., 2023; Kurmanji et al., 2023) rely heavily on gradients, leading to limitations in computational cost and samples required, especially for large models and datasets. Our work addresses these limitations by introducing a novel and demonstrably superior solution for class-wise unlearning, outperforming state-of-the-art baselines. Our approach is radically different most of the current approaches in the following ways:

1. **Gradient-Free:** We eliminate the computational burden and potential instability of gradient based unlearning.
2. **Single-Step Weight Update:** Our method achieves unlearning in a single update, similar to Foster et al. (2024), and surpasses the iterative nature of many baselines.
3. **Novel Weight Update:** Our SVD-driven activation suppression mechanism stands out from traditional update strategies.
4. **Efficiency Benefits:** Our proposed algorithm is sample efficient while having low runtimes on large datasets like ImageNet.

We demonstrate our algorithm on a simple 4-way classification problem with input containing 2 features as shown in Figure 1. The decision boundary learned by the trained model is shown in Figure 1(a) while the model unlearning the red class with our method exhibits the decision boundary depicted in Figure 1(b). The decision boundary for a retrained model is shown if Figure 1(c). This illustration shows that the proposed algorithm redistributes the input space of the unlearned class to the closest classes. See Appendix A.1 for experimental details.

In summary, the contributions of this work are as follows:

- We propose a novel Singular Value Decomposition based class unlearning algorithm which (a) is Gradient-Free, (b) uses single step weight update, (c) has novel weight update mechanism, (d) is compute and sample efficient and (e) improves the unlearning efficacy as compared to SoTA algorithms. To the best of our knowledge, our work is the first to demonstrate class unlearning results on ImageNet for SoTA transformer based models.
- Our algorithm exhibits competitive unlearning performance and resilience against Membership Inference Attacks (MIA). Compared to baselines, it achieves an average accuracy improvement of 1.38% on the ImageNet dataset while requiring up to $10\times$ fewer samples for unlearning and maintaining competitive runtimes. Additionally, under stronger MIA attacks on the CIFAR-100 dataset using a ResNet18 architecture, our approach outperforms the best baseline by 1.8%. Further, we provide evidence that our model's behavior aligns with that of a model retrained without the forget class samples through membership inference attacks, saliency-based feature analyses, and confusion matrix analyses. Additionally, we extend our approach to multi-class unlearning or task unlearning to demonstrate the capability of processing multiple unlearning requests.

## 2 Related Works

**Unlearning:** Many unlearning algorithms have been introduced in the literature, addressing various unlearning scenarios, including item unlearning (Bourtoule et al., 2021), feature unlearning (Warnecke et al., 2021), class unlearning (Tarun et al., 2023), and task unlearning (Parisi et al., 2019). Some of these solutions make simplifying assumptions on the learning algorithm. For instance, Ginart et al. (2019) demonstrate unlearning within the context of k-mean clustering, Brophy & Lowd (2021) present their algorithm for random forests, Mahadevan & Mathioudakis (2021) and Izzo et al. (2021) propose an algorithm in the context of linear/logistic regression. Further, there have been efforts in literature, to scale these algorithms for convolution layers Golatkar et al. (2020a;b). Note, however, the algorithms have been only demonstrated on small scale problems. In contrast, other works, such as Bourtoule et al. (2021), suggest altering the training process to enable efficient unlearning. This approach requires saving multiple snapshots of the model from different stages of training and involves retraining the model for a subset of the training data, effectively trading off compute and memory for good accuracy on retain samples. Unlike these works our proposed algorithm does not make any assumptions on the training process or the algorithm used for training the original model. Recent work on Exact Unlearning (EU-k) and Catastrophic Forgetting (CF-k) proposed in Goel et al. (2022) presents fine-tuning methods that update the last k layers of the network. EU-k re-initializes the last k layers and trains on the retain set, while CF-k directly fine-tunes the last k layers, leveraging learning dynamics for unlearning. Further, a few-shot unlearning approach proposed by Yoon et al. (2022) trains a generative model to create proxy data for the retain-set and forget-set. This algorithm unlearns by fine-tuning on the concatenated retain-set and relabeled forget-set. These algorithms are evaluated on small datasets and have high compute requirements, possibly restricting their use for large datasets. Selectively Synaptic Dampening (SSD) (Foster et al., 2024) introduces a method to unlearn using the Fisher information. Like our method, this approach enables unlearning in a single update step; however, it is shown to require a large dataset size Goel et al. (2024).

**Class Unlearning:** The current state-of-the-art (SoTA) for class unlearning is claimed by Tarun et al. (2023). In their work, the authors propose a three stage unlearning process, where the first stage learns a noise pattern that maximizes the loss for each of the classes to be unlearned. The Second stage (the impair stage) unlearns the class by mapping the noise to the forget class. Finally, as the impair stage is seen to reduce the accuracy on the retained classes, the authors propose to finetune the impaired model on the subset of training data in the third stage (the repair stage). This work presents the results on small datasets with undertrained models and utilizes up to 20% of the training data for the unlearning process. Further, the work by Chundawat et al. (2023) proposes two algorithms which assumes no access to the training samples. Additionally, authors of Baumhauer et al. (2022) propose a linear filtration operator to shift the classification of samples from unlearn class to other classes. These works lose considerable accuracy on the retain class samples and have been demonstrated on small scale datasets like MNIST and CIFAR10. Our work demonstrates results on SoTA vision transformer models for the ImageNet dataset, showing the effective scaling of our algorithm on large dataset with the model trained to convergence.

**Other Related Algorithms :** SVD is used to constrain the learning in the direction of previously learned tasks in the continual learning setup Saha et al. (2021); Chen et al. (2022); Saha & Roy (2023). These methods are sample efficient in estimating the gradient space relevant to a task. Recent work by Li et al. (2023) proposes subspace based federated unlearning using SVD. The authors perform gradient ascent in the orthogonal space of input gradient spaces formed by other clients to eliminate the target client's contribution in a federated learning setup. Such ascent based unlearning is generally sensitive to hyperparameters and is susceptible to catastrophic forgetting on retain samples. As our proposed approach does not rely on such gradient based training steps it is less sensitive to the hyperparameters. Moreover, such techniques could be used on top of our method to further enhance the unlearning performance.

## 3 Preliminaries

**Unlearning:** Let the training dataset be denoted by $\mathcal{D}_{train} = \{(x_i, y_i)\}_{i=1}^{N_{train}}$ consisting of $N_{train}$ training samples where $x_i$ represents the network input and $y_i$ is the corresponding label. The test dataset be $\mathcal{D}_{test} = \{(x_i, y_i)\}_{i=1}^{N_{test}}$ containing $N_{test}$ samples. Consider a function $y = f(x_i, \theta)$ with the parameters $\theta$ that

approximates the mapping of the inputs $x_i$ to their respective labels $y_i$. A well-trained deep learning model with parameters $\theta$, would have numerous samples $(x_i, y_i) \in \mathcal{D}_{test}$ for which the relationship $y_i = f(x_i, \theta)$ holds. For a class unlearning task aimed at removing the target class $t$, the training dataset can be split into two partitions, namely the retain dataset $\mathcal{D}_{train\_r} = \{(x_i, y_i)\}_{i=1;y_i \neq t}^{N}$ and the forget dataset $\mathcal{D}_{train\_f} = \{(x_i, y_i)\}_{i=1;y_i=t}^{N}$. Similarly, the test dataset can be split into these partitions as $\mathcal{D}_{test\_r} = \{(x_i, y_i)\}_{i=1;y_i \neq t}^{N}$ and $\mathcal{D}_{test\_f} = \{(x_i, y_i)\}_{i=1;y_i=t}^{N}$. The objective of the class unlearning algorithm is to derive unlearned parameters $\theta_f^*$ based on $\theta$, a subset of the retain partition $\mathcal{D}_{train\_sub\_r} \subset \mathcal{D}_{train\_r}$, and a subset of the forget partition $\mathcal{D}_{train\_sub\_f} \subset \mathcal{D}_{train\_f}$. The parameters $\theta_f^*$ must be functionally indistinguishable from a network with parameters $\theta^*$, which is retrained from scratch on the samples of $\mathcal{D}_{train\_r}$ in the output space. Empirically, these parameters must satisfy $f(x_i, \theta^*) \simeq f(x_i, \theta_f^*)$ for $(x_i, y_i) \in \mathcal{D}_{test}$.

**SVD:** A rectangular matrix $A \in \mathbb{R}^{d \times n}$ can be decomposed using SVD as $A = U\Sigma V^T$ where $U \in \mathbb{R}^{d \times d}$ and $V \in \mathbb{R}^{n \times n}$ are orthogonal matrices and $\Sigma \in \mathbb{R}^{d \times n}$ is a diagonal matrix containing singular values Deisenroth et al. (2020). The columns of matrix $U$ are the $d$ dimensional orthogonal vectors sorted by the amount of variance they explain for $n$ samples (or the columns) in the matrix $A$. These vectors are also called the basis vectors explaining the column space of A. For the $i^{th}$ vector in $U$, $u_i$, the amount of the variance explained is proportional to the square of the $i^{th}$ singular value $\sigma_i^2$. Hence the percentage variance explained by a basis vector $u_i$ is given by $\sigma_i^2/(\sum_{j=1}^{d}(\sigma_j^2))$.

## 4 Unlearning Algorithm

Algorithm 1 presents the pseudocode of our approach. Consider $l^{th}$ linear layer of a network given by $x_o^l = x_i^l(\theta^l)^T$, where $\theta^l$ signifies the parameters, $x_i^l$ stands for input activations and $x_o^l$ denotes output activation. Our algorithm aims to suppress the class-discriminatory activations associated with forget class. When provided with a class-discriminatory projection matrix $P_{dis}^l$, suppressing this class-discriminatory activation from the input activations gives us $x_i^l - x_i^l P_{dis}^l$. Post multiplying the parameters with $(I - P_{dis}^l)^T$, is mathematically the same as removing class-discriminatory information from $x_i^l$ as shown in Equation 1. This enables us to modify the model parameters to destroy the class-discriminatory activations given the matrix $P_{dis}^l$ and is accomplished by the `update_parameter()` function in line 16 of pseudocode presented in Algorithm 1. The next Subsection focuses on optimally computing this class-discriminatory projection matrix, $P_{dis}^l$.

$$x_o^l = \underbrace{(x_i^l - x_i^l P_{dis}^l)}_{\text{Activation Suppression}} (\theta^l)^T = x_i^l(I - P_{dis}^l)(\theta^l)^T = x_i^l(\underbrace{\theta^l(I - P_{dis}^l)^T}_{\text{updated parameter}})^T \tag{1}$$

### 4.1 Class Discriminatory Space

We need to estimate the activation space for retain class samples and the forget class samples before computing $P_{dis}$. These spaces, also referred to as the Retain Space ($U_r^l$) and the Forget Space ($U_f^l$), are computed using SVD on the accumulated representations for the corresponding samples as elaborated in Section 4.1.1. Subsequently, we scale the basis vectors within their respective spaces using importance scores proportional to the variance explained by each vector. This yields the scaled retain projection space $P_r^l$ and the scaled forget projection space $P_f^l$. Finally, we compute the shared information between these spaces and remove it from the scaled forget space to isolate class-discriminatory feature space as explained in Subsection 4.1.2.

#### 4.1.1 Space Estimation via SVD on Representations

This Subsection provides the details on Space Estimation given by lines 3-7 of Algorithm 1. We give an in-depth explanation for obtaining the Retain Space, $U_r$. We use a small subset of samples from the classes to be retained, denoted as $X_r = \{x_i\}_{i=1}^{K_r}$, where $K_r$ represents the number of retain samples. Forget Space ($U_f$) is estimated similarly on the samples from the class to be unlearned, denoted by $X_f = \{x_i\}_{i=1}^{K_f}$ where $K_f$ represents the number of samples.

**Representation Matrix:** To obtain the basis vectors for the Retain Space through SVD, we collect the representative information (activations) for each layer in the network and organize them into a representation

---

**Algorithm 1** Proposed Unlearning Algorithm

---

**Input:** $\theta$ is the parameters of the original model; $X_r$ and $X_f$ is small set of inputs $x_i$ sampled independently from $\mathcal{D}_{train\_r}$ and $\mathcal{D}_{train\_f}$ respectively; $\mathcal{D}_{train\_sub\_r} \subset \mathcal{D}_{train\_r}$; $\mathcal{D}_{train\_sub\_f} \subset \mathcal{D}_{train\_f}$; and alpha\_r\_list and alpha\_f\_list are list of hyperparameters $\alpha_r$ and $\alpha_f$ respectively.

**procedure Unlearn(** $\theta$, $X_r$, $X_f$, $\mathcal{D}_{train\_sub\_r}$, $\mathcal{D}_{train\_sub\_f}$, alpha\_r\_list, alpha\_f\_list **)**

1. best\_score = **get\_score**($\theta$, $\mathcal{D}_{train\_sub\_r}$, $\mathcal{D}_{train\_sub\_f}$)
2. $\theta_f^* = \theta$
3. $R_r = $ **get\_representation**(model, $X_r$)
4. $R_f = $ **get\_representation**(model, $X_f$)
5. **for** each linear and convolution layer $l$ **do**
6.     $U_r^l$, $\Sigma_r^l = $ **SVD**($R_r^l$)
7.     $U_f^l$, $\Sigma_f^l = $ **SVD**($R_f^l$)
8. **for** each $\alpha_r \in$ alpha\_r\_list **do**
9.    **for** each $\alpha_f \in$ alpha\_f\_list **do**
10.    **for** each linear and convolution layer $l$ **do**
11.      $\Lambda_r^l = $ **scale\_importance**($\Sigma_r^l$, $\alpha_r$)
12.      $\Lambda_f^l = $ **scale\_importance**($\Sigma_f^l$, $\alpha_f$)
13.      $P_r^l = U_r^l \Lambda_r^l$ **transpose**($U_r^l$)
14.      $P_f^l = U_f^l \Lambda_f^l$ **transpose**($U_f^l$)
15.      $P_{dis}^l = P_f^l(I - P_r^l)$
16.      $\theta_f^l = $ **update\_parameter**($I - P_{dis}^l$, $\theta^l$ )
17.    score = **get\_score**($\theta_f$, $\mathcal{D}_{train\_sub\_r}$, $\mathcal{D}_{train\_sub\_f}$)
18.    **if** score > best\_score **do**
19.     best\_score = score;     $\theta_f^* = \theta_f$

**return** $\theta_f^*$

---

matrix denoted as $R_r^l$. We accumulate these representation matrices, $R_r^l$, for both linear and convolutional layers in a list $R_r = [R_r^l]_{l=1}^L$, where $L$ is the number of layers. For the linear layer, this matrix is given by $R_r^l = [(x_i^l)_{i=1}^{K_r}]$, which stacks the input activations $x_i^l$ to obtain a matrix of size $K_r \times d^l$, where $d^l$ is the input dimension of $l^{th}$ linear layer.

A convolutional layer has to be represented as a matrix multiplication to apply the proposed weight update rule. This is done using the unfold operation (Liu et al., 2018) on input activation maps. Consider a convolutional layer with $C_i$ input channels and $k$ as kernel size with the output activation, $x_o^l$, having the resolution of $h_o \times w_o$, where $h_o$ and $w_o$ is the height and width of the output activations. There are $h_o w_o$ patches of size $C_i \times k \times k$ in the input activation $x_i^l$ on which the convolution kernel operates to obtain the values at the corresponding locations in the output map. The unfold operation flattens each of these $h_o w_o$ patches to get a matrix of size $h_o w_o \times C_i k k$. Now, if we reshape the weight as $C_i k k \times C_o$ where $C_o$ is the output channels, we see that the convolution operation becomes a matrix multiplication between the unfolded matrix and the reshaped weights achieving the intended objective. This is graphically presented in Figure 1 of Saha et al. (2021). Hence, the representation matrix for the convolutional layer is given by $R_r^l = [(\texttt{unfold}(x_i^l)^T)_{i=1}^{K_r}]$. The representation matrix is obtained by `get_representation()` function in lines 3-4 of the Algoithm 1.

**Space Estimation:** We perform SVD on the representation matrices for each layer as shown in lines 6-7 of the Algorithm 1. `SVD()` function returns the basis vectors $U_r^l$ that span the activation of the retain samples in $X_r$ and the singular values $\Sigma_r^l$ for each layer $l$. Retain Space $U_r = [U_r^l]_{l=1}^L$ is the list of these basis vectors for all the layers.

### 4.1.2 Projection Space

The basis vectors in the Spaces obtained are orthonormal and do not capture any information about the importance of the basis vector. The information of the significance of the basis vector is given by the

corresponding singular value. Hence, we propose to scale the basis vector in proportion to the amount of variance they explain in the input space as presented below.

**Importance Scaling:** To capture the importance the $i^{th}$ basis vector in the matrix $U$ (or the $i^{th}$ column of $U$), we formulate an diagonal importance matrix $\Lambda$ having the $i^{th}$ diagonal component $\lambda_i$ given by Equation 2, where d is the number of basis vectors.. Here $\sigma_i$ represents the the $i^{th}$ singular value in the matrix $\Sigma$. The parameter $\alpha \in (0, \infty)$ called the scaling coefficient is a hyperparameter that controls the scaling of the basis vectors. When $\alpha$ is set to 1 the basis vectors are scaled by the amount of variance they explain. As $\alpha$ increases the importance score for each basis vector increases and reaches 1 as $\alpha \to \infty$. Similarly, a decrease in $\alpha$ decreases the importance of the basis vector and goes to 0 as $\alpha \to 0$. This operation is represented by `scaled_importance()` function in lines 11-12 of the Algorithm 1. It is important to note that without the proposed scaling approach the matrices $P_r$ and $P_f$ become identity matrices in lines 13-14 of Algorithm 1, as $U$ is an orthonormal matrix. This in turn makes $P_{dis}$ a zero matrix, which means the weight update in line 16 of Algorithm 1 projects weight on the identity matrix mathematically restricting unlearning. Hence it is important to use scaling in lines 11-12 of Algorithm 1.

$$\lambda_i = \frac{\alpha \sigma_i^2}{(\alpha - 1)\sigma_i^2 + \sum_{j=1}^{d} \sigma_j^2} \tag{2}$$

**Scaled Projection Matrix ($P$):** Given the scaling coefficient $\alpha_r$ and $\alpha_f$, we compute the importance scaling matrix $\Lambda_r$ and $\Lambda_f$ as per Equation 2. The scaled retain projection matrix, which projects the input activations to the retain space is given by $P_r^l = U_r^l \Lambda_r^l (U_r^l)^T$ and the scaled forget projection matrix given by $P_f^l = U_f^l \Lambda_f^l (U_f^l)^T$, see line 13-14 in Algorithm 1.

**Class-Discriminatory Projection Matrix ($P_{dis}$):** We obtain the class discriminatory projection matrix, $P_{dis}$, by removing the shared space given by $P_f^l P_r^l$ from the forget projection matrix. Mathematically, this can also be written as $P_{dis}^l = P_f^l - P_f^l P_r^l = P_f^l (I - P_r^l)$. Intuitively, this projects the forget space onto the space that does not contain any information about the retain space, effectively removing the shared information from the forget projection matrix to obtain $P_{dis}$. Our algorithm introduces two hyperparameters namely $\alpha_r$ and $\alpha_f$ the scaling coefficients for the Retain Space and the Forget Space respectively. Subsection 4.2 presents a discussion on tuning these hyperparameters.

## 4.2 Hyperparameter Search

**Grid Search:** As seen in lines 8-9 of the Algorithm 1 we do a grid search over all the possible values of $\alpha_r$ and $\alpha_f$ provided in *alpha_r_list* and *alpha_f_list* to obtain the best unlearned parameters $\theta_f^*$. We observe this search is necessary for our algorithm. One intuitive explanation for this is that the unlearning class may exhibit varying degrees of confusion with the retain classes, making it easier to unlearn some classes compared to others, hence requiring different scaling for the retain and forget spaces. Note, we observe that increasing the value of $\alpha_f$ decreases the retain accuracy $acc_r$ and hence we terminate the inner loop (line 9) to speed up the grid search and have not presented this in the Algorithm 1 for simplicity. We introduce a simple scoring function given below to rank the unlearned models with different pairs of $\alpha_r$ and $\alpha_f$.

$$score = acc_r(1 - acc_f/100) \tag{3}$$

**Score:** The proposed scoring function given by Equation 3 returns penalized retain accuracy, where $acc_r$ and $acc_f$ are the accuracy on the $\mathcal{D}_{train\_sub\_r}$ and $\mathcal{D}_{train\_sub\_f}$ respectively. This function returns a low score for an unlearned model having a high value of $acc_f$ or an unlearned model having low $acc_r$. The value of the returned score is high only when $acc_r$ is high and $acc_f$ is low, which is the desired behavior of the unlearned model.

## 4.3 Discussion

**Compute and Sample Efficiency:** The speed and efficiency of our approach can be attributed to the design choices. Firstly our method runs inference for very few samples to get the representations $R$. This in turn results in small size of the representation matrices ensuring the SVD is fast and computationally cheap.

Additionally, the SVD operation for each layer is independent and can be parallelized to further improve speeds. Secondly, our approach only performs inference and does not rely on computationally intensive gradient based optimization steps (which also require tuning the learning rates) and gets the unlearned model in a single step for each grid search (over $\alpha_r$ and $\alpha_f$) leading to a fast and efficient approach. Additionally, our approach has fewer hyperparameters to tune compared to gradient-ascent based baselines, which are sensitive to choices of optimizer, learning rate, batch size, learning rate scheduler, weight decay, etc.

**Scaling to Transformer Architecture:** We can readily extend proposed approach to transformer architectures by applying our algorithm to all the linear layers in the architecture. Note, that we do not change the normalization layers for any architecture as the fraction of total parameters for these layers is insignificant (see Table 5 in Appendix A.2).

**Limitations:** Our algorithm assumes a significant difference in the distribution of forget and retain samples for SVD to find distinguishable spaces from a few random samples. This is true in class unlearning setup, where retain and forget samples come from different non-overlapping classes. However, in the case of unlearning a random subset of training data (Kurmanji et al., 2023), we would require additional modification for effective unlearning. Further, machine unlearning is an evolving field, and current evaluation techniques can be misled by naive baselines, such as setting the bias term to a large negative number or hard-coding the network to output nothing (or a random class) if the prediction is the forget class. These trivial baselines might satisfy certain metrics (e.g., accuracy-based metrics) but fail others (e.g., specific types of attacks), as information about the forget set classes can still persist in the trained weights. This suggests that existing evaluations for unlearning may be insufficient to guarantee privacy . We hope future works will address these shortcomings in evaluating the privacy provided by unlearning algorithms.

## 5 Experiments

**Dataset and Models :** We conduct the class unlearning experiments on CIFAR10, CIFAR100 (Krizhevsky et al., 2009) and ImageNet (Deng et al., 2009) datasets. We use the modified versions of ResNet18 (He et al., 2016) and VGG11 (Simonyan & Zisserman, 2014) with batch normalization for the CIFAR10 and CIFAR100 datasets. The training details for these models are provided in Appendix A.3. For the ImageNet dataset, we use the pretrained VGG11 and base version of Vision Transformer with a patch size of 14 (Dosovitskiy et al., 2020) available in the torchvision library.

**Comparisons:** We benchmark our method against 5 unlearning approaches. Three of these approaches Retraining, NegGrad and NegGrad+ are common baselines used in literature such as (Tarun et al., 2023; Kurmanji et al., 2023). Retraining involves training the model from scratch using the retain partition of the training set, $D_{train\_r}$, and serves as our goldstandard model. A detailed explanation of NegGrad and NegGrad+ with psuedocodes is presented in Appendix A.4 and A.5 respectively. Further, we compare our work with recent algorithms (Tarun et al., 2023; Kurmanji et al., 2023; Foster

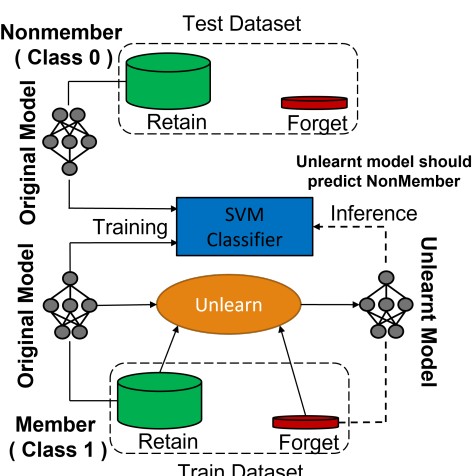

**Figure 2:** Membership Inference Attack.

et al., 2024) to demonstrate the effectiveness of our approach. Discussion on hyperparameters is presented in Appendix A.6.

**Evaluation:** In our experiments we evaluate the model with the accuracy on retain samples $ACC_r$ and the accuracy on the forget samples $ACC_f$. In addition, we implement Membership Inference Attack ($MIA$) to distinguish between samples in $\mathcal{D}_{train\_r}$ (Member class) and $\mathcal{D}_{test\_r}$ (Nonmember class) as shown in Figure 2. A strong MIA attack is proposed by Hayes et al. (2024), however, this attack requires substantial computational resources and are not practical for evaluating large datasets on large models hence we evalutated the unlearning on simple MIA. We use the confidence scores for the target class and train a Support Vector

**Table 1:** Single class Forgetting on CIFAR dataset. We bold font the method with highest value for $ACC_r \times (100 - ACC_f) \times MIA$.

| | Method | VGG11_BN | | | ResNet18 | | |
|---|---|---|---|---|---|---|---|
| | | $ACC_r(\uparrow)$ | $ACC_f(\downarrow)$ | $MIA(\uparrow)$ | $ACC_r(\uparrow)$ | $ACC_f(\downarrow)$ | $MIA(\uparrow)$ |
| CIFAR10 | Original | $91.58 \pm 0.52$ | $91.58 \pm 4.72$ | $0.11 \pm 0.08$ | $94.89 \pm 0.31$ | $94.89 \pm 2.75$ | $0.03 \pm 0.03$ |
| | Retraining | $92.58 \pm 0.83$ | $0$ | $100 \pm 0$ | $94.81 \pm 0.52$ | $0$ | $100 \pm 0$ |
| | NegGrad | $81.46 \pm 5.67$ | $0.02 \pm 0.04$ | $0$ | $69.89 \pm 10.23$ | $0$ | $0$ |
| | NegGrad+ | $89.79 \pm 1.49$ | $0.13 \pm 0.16$ | $99.93 \pm 0.15$ | $89.91 \pm 1.41$ | $0.94 \pm 1.87$ | $98.68 \pm 1.42$ |
| | Tarun et al. (2023) | $89.21 \pm 0.84$ | $0$ | $0$ | $92.20 \pm 0.72$ | $10.89 \pm 8.79$ | $61.5 \pm 25.86$ |
| | Kurmanji et al. (2023) | $92.09 \pm 0.89$ | $0$ | $0$ | $94.79 \pm 0.63$ | $0$ | $0$ |
| | Foster et al. (2024) | $63.23 \pm 31.87$ | $18.28 \pm 35.90$ | $29.21 \pm 47.08$ | $85.76 \pm 25.76$ | $4.37 \pm 12.795$ | $87.86 \pm 31.21$ |
| | Ours | $\mathbf{91.77 \pm 0.69}$ | $\mathbf{0}$ | $\mathbf{98.28 \pm 5.43}$ | $\mathbf{94.19 \pm 0.50}$ | $\mathbf{0.03 \pm 0.09}$ | $\mathbf{95.5 \pm 14.23}$ |
| CIFAR100 | Original | $69.22 \pm 0.29$ | $67 \pm 15.23$ | $0.2 \pm 0.25$ | $76.64 \pm 0.13$ | $74.3 \pm 13.27$ | $0.08 \pm 0.1$ |
| | Retraining | $68.97 \pm 0.40$ | $0$ | $100 \pm 0$ | $76.81 \pm 0.50$ | $0$ | $100 \pm 0$ |
| | NegGrad | $51.21 \pm 6.37$ | $0$ | $0$ | $60.32 \pm 7.03$ | $0$ | $29.98 \pm 48.27$ |
| | NegGrad+ | $58.66 \pm 3.91$ | $0$ | $0$ | $71.37 \pm 2.78$ | $0$ | $100 \pm 0$ |
| | Tarun et al. (2023) | $53.94 \pm 1.22$ | $0$ | $0$ | $63.39 \pm 0.50$ | $3.1 \pm 5.65$ | $0$ |
| | Kurmanji et al. (2023) | $69.09 \pm 0.30$ | $0$ | $0$ | $76.40 \pm 0.19$ | $0$ | $0$ |
| | Foster et al. (2024) | $47.433 \pm 26.197$ | $0$ | $20 \pm 42.16$ | $\mathbf{76.52 \pm 0.23}$ | $\mathbf{1.6 \pm 5.06}$ | $\mathbf{100 \pm 0}$ |
| | Ours | $\mathbf{65.94 \pm 1.21}$ | $\mathbf{0.3 \pm 0.48}$ | $\mathbf{99.92 \pm 0.10}$ | $73.60 \pm 1.41$ | $0.3 \pm 0.48$ | $100 \pm 0$ |

**Table 2:** Single class forgetting on ImageNet-1k dataset.

| Method | Retain samples | VGG11_BN | | | ViT_B_16 | | |
|---|---|---|---|---|---|---|---|
| | | $ACC_r(\uparrow)$ | $ACC_f(\downarrow)$ | $MIA(\uparrow)$ | $ACC_r(\uparrow)$ | $ACC_f(\downarrow)$ | $MIA(\uparrow)$ |
| Original | - | $68.61 \pm 0.02$ | $72.6 \pm 25.92$ | $22.72 \pm 22.59$ | $80.01 \pm 0.037$ | $80.6 \pm 19.87$ | $13.36 \pm 12.94$ |
| NegGrad+ | - | $66.37 \pm 1.27$ | $8.8 \pm 11.48$ | $96.58 \pm 4.40$ | $73.76 \pm 1.46$ | $0$ | $99.98 \pm 0.05$ |
| Tarun et al. (2023) | 9990 | $43.5618 \pm 0.59$ | $0$ | $98.96 \pm 3.26$ | $56.00 \pm 3.47$ | $38.8 \pm 34.074$ | $66.67 \pm 50$ |
| Kurmanji et al. (2023) | 9990 | $\mathbf{67.29 \pm 0.34}$ | $\mathbf{0}$ | $\mathbf{99.92 \pm 0.15}$ | $78.675 \pm 0.13$ | $2.0 \pm 4.0$ | $0$ |
| Foster et al. (2024) | 9990 | $64.39 \pm 6.14$ | $0$ | $60 \pm 51.64$ | $\mathbf{79.93 \pm 0.08}$ | $\mathbf{0.2 \pm 0.63}$ | $\mathbf{99.8 \pm 0.01}$ |
| Foster et al. (2024) | 999 | $29.687 \pm 19.57$ | $9.6 \pm 21.41$ | $0$ | $55.03 \pm 26.55$ | $20.2 \pm 30.03$ | $20 \pm 42.16$ |
| Ours | $\mathbf{999}$ | $66.41 \pm 0.60$ | $0.6 \pm 1.35$ | $99.33 \pm 0.90$ | $78.47 \pm 0.84$ | $0.2 \pm 0.63$ | $99.98 \pm 0.05$ |

Machine (SVM) (Hearst et al., 1998) classifier. In our experiments, $MIA$ scores represent the average model prediction accuracy for $\mathcal{D}_{train\_f}$ classified as Nonmember. A high value of $MIA$ score for a given model indicates the failure of MIA model to detect $\mathcal{D}_{train\_f}$ as a part of training data. An unlearned model is expected to match the $MIA$ score of the Retrained model. See Appendix A.7 for more details on MIA setup.

## 6 Results and Analyses

**Class Forgetting:** We present the results for single class forgetting in Table 1 for the CIFAR10 and CIFAR100 dataset. The table presents results that include both the mean and standard deviation across 10 different target unlearning classes. CIFAR10 dataset is accessed for unlearning on each class and CIFAR100 is evaluated for every 10th starting from the first class. The Retraining approach matches the accuracy of the original model on retain samples and has 0% accuracy on the forget samples, which is the expected upper bound. The $MIA$ accuracy for this model is 100% which signifies that MIA model is certain that $\mathcal{D}_{train\_f}$ does not belong to the training data. The NegGrad method shows good forgetting with low $ACC_f$, however, performs poorly on $ACC_r$ and $MIA$ metrics. The NegGrad algorithm's performance on retain samples is expected to be poor because it lacks information about the retain samples required to protect the relevant features. Further, this explains why NegGrad+ which performs gradient descent on the retain samples along with NegGrad can maintain impressive performance on retain accuracy with competitive forget accuracy. In some of our experiments, we observe that the NegGrad+ approach outperforms the SoTA benchmarks (Foster et al., 2024; Tarun et al., 2023) and (Kurmanji et al., 2023) which suggests the NegGrad+ approach is a strong baseline for class unlearning. For CIFAR datasets, our proposed training free algorithm achieves a better tradeoff between the evaluation metrics when compared against all the baselines, with an exception of ResNet18 on CIFAR100 where Foster et al. (2024) performs better. Further, we observe the MIA numbers

for our method close to the retrained model and better than all the baselines for most of our experiments. We demonstrate our algorithm easily scales to ImageNet without compromising its effectiveness, as seen in Table 2. Due to the training complexity of the experiments, we were not able to obtain retrained models for ImageNet. We observe that the results on CIFAR10 and CIFAR100 datasets consistently show $ACC_f$ to be 0 and the $MIA$ performance to be 100%. We, therefore, interpret the model with high $ACC_r$, $MIA$, and low $ACC_f$ as a better unlearned model for these experiments. We conduct unlearning experiments on the ImageNet dataset for every $100^{th}$ class starting from the first class. Our algorithm shows less than 1.5% drop in $ACC_r$ as compared to the original model while maintaining less than 1% forget accuracy for a well trained SoTA Transformed based model. We present the results on varients of ViT model in Appendix A.8 The MIA scores for our model are nearly 100% indicating that model the MIA model fails to recognize $\mathcal{D}_{train\_f}$ as part of training data. We observe that (Kurmanji et al., 2023) outperforms our method for a VGG11 model trained on ImageNet, however, requires access to $10\times$ more retain samples and computationally expensive than our approach. We observe Foster et al. (2024) with 9990 retain samples outperforms our method, however, reducing the number of retain samples to be comparable to our method lead to a sever decrease in the performance. Further, we also explore 2 alternative variants of our algorithm in Appendix A.9. Due to their inferior performance in terms of $ACC_r$ and $ACC_f$, we decided not to further evaluate them in this work.

**Strong MIA evaluation:** We evaluate our method under stronger MIA as recomended by Hayes et al. (2024). We trained eight models on the entire CIFAR100 training dataset using ResNet18 with the same hyperparameters as the original training. These models act as the shadow models for unlearning. We obtained the shadow unlearnt model by applying the unlearning algorithms to these eight models. Next, we trained eight models without the target forget-class in the training dataset, referred to as the shadow retrained models. Following the recommendations by (Hayes et al., 2024), we applied various unlearning techniques to these eight models to unlearn the target class. We then trained the MIA model to distinguish between forget samples in the shadow unlearnt model and the shadow retrained model (without the forget-set). Specifically, we used seven of these models to learn the distributions (mean and standard deviations) and used the eighth model pair to determine the detection threshold. Finally, we tested the unlearnt models on ResNet18 for CIFAR100 dataset from Table 1 of our paper using this U-LIRA MIA attack.

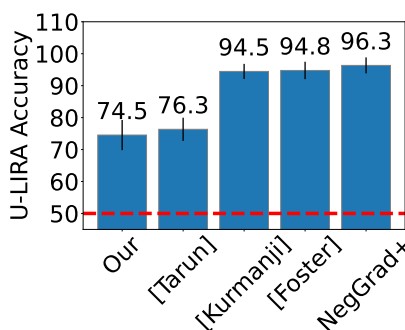

**Figure 3:** U-LIRA Hayes et al. (2024) Membership Inference Attack for CIFAR10 dataset on ResNet18 model.

We used the forget set from the original model as specified by (Hayes et al., 2024) and reported the accuracy of the MIA attack in identifying samples as belonging to the forget-set. We conducted this analysis for all the target classes used in Table 1. Ideally, this accuracy should be close to 50%, indicating that the MIA attack fails to distinguish between the forget-set from the unlearnt and the retrained model (implying perfect class unlearning). The results in Figure 3, shows our algorithm outperforms the SoTA under a strong MIA.

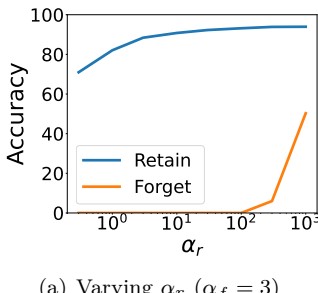

(a) Varying $\alpha_r$ ($\alpha_f = 3$)

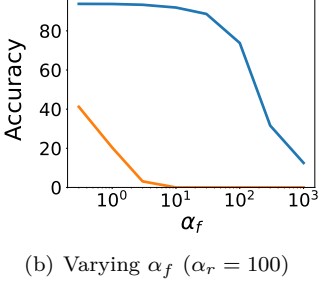

(b) Varying $\alpha_f$ ($\alpha_r = 100$)

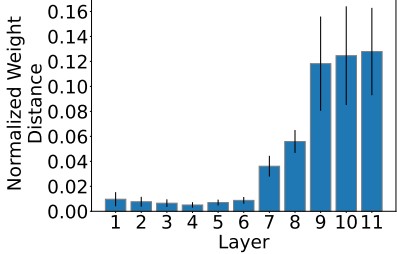

**Figure 5:** Layer-wise weight change for VGG11 on CIFAR10 dataset.

**Figure 4:** Effect of varying (a) $\alpha_r$ and (b) $\alpha_f$ for Cat class of CIFAR10 dataset on VGG11 network.

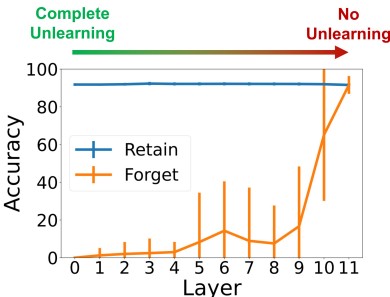

**Figure 6:** Effect of applying unlearning to the later layers for VGG11 model on CIFAR10 dataset.

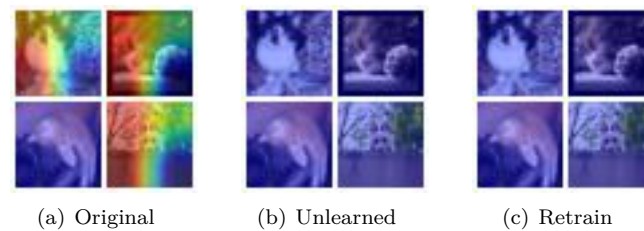

(a) Original     (b) Unlearned     (c) Retrain

**Figure 7:** GradCAM-based heatmaps for (a) Original, (b) Retrained, and (c) Unlearned VGG11 model on the CIFAR10 with a cat as the target class, demonstrating that the unlearned model does not highlight any features specific to the cat. we provide more saliency maps in Appendix A.11

**Effect of $\alpha$:** Figure 4 shows the impact of varying $\alpha_r$ and $\alpha_f$ for unlearning the cat class from CIFAR10 dataset using VGG11 network. In Figure 4(a), we set $\alpha_f = 3$ and vary $\alpha_r$ in $[0.3, 1, 3, 10, 30, 100, 300, 1000]$. Similarly, in Figure 4(b), we keep $\alpha_r = 100$ and vary $\alpha_f$ in $[0.3, 1, 3, 10, 30, 100, 300, 1000]$. Note, here $\alpha_r = 100$ and $\alpha_f = 3$ are the optimal hyperparameter for unlearning the cat class. We observe that $ACC_r$ increases with increase in $\alpha_r$ and decreases with increase in $\alpha_f$. Similarly, $ACC_f$ increases with increase in $\alpha_r$ and decreases with increase in $\alpha_f$, however, the $ACC_f = 0$ for $\alpha_r < 100$ and $\alpha_f > 1$. We observed similar trends for other classes. These trends can be understood by analysing the overall update equation of our algorithm (combining lines 11-16 ) given by Equation 4 (see Appendix A.10 for the explanation).

$$\theta_f = (I - P_f(I - P_r))\theta \tag{4}$$

**Layer-wise analysis:** We plot the layerwise weight difference between the parameters of the unlearned and the original model for VGG11 on the CIFAR10 dataset in Figure 5. We observe that the weight change is larger for the later layers. This suggests that the class discriminatory features are more prominent in the later layers of the network which is consistent with the findings of Olah et al. (2017). In Figure 6, we plot $ACC_r$ and $ACC_f$ when our algorithm is applied to top few layers for CIFAR10 dataset on VGG11 model. The x-axis in the plot represents the number of initial layers $n$ we do not apply the unlearning algorithm. A value of $n$ on x-axis represents a case where we do not change the initial layers $0 - n$(including $n$) in unlearning and apply our unlearning algorithm to the rest of the layers. We observe that the effect of removing the projections is minimal on $ACC_r$. The forget accuracy keeps increasing as we sequentially remove the projections starting from the initial layers. We observe that applying our algorithm to all the layers ensures low mean and standard deviation for $ACC_f$.

**Saliency-based analysis :** We test this VGG11 model unlearning the cat class with GradCAM-based feature analysis as presented in Figure 7, and we see that our model is unable to detect class discriminatory information which validates unlearning. This behavior is similar to the retrained VGG11 model shown in Figure 7(c).

**Confusion Matrix analysis:** We plot the confusion matrix showing the distribution of true labels and predicted labels for the original VGG11 model and VGG11 model unlearning cat class with our algorithm for CIFAR10 in Figure 8. Interestingly, we observe that a significant portion of the cat samples are redistributed across the animal categories. The majority of these samples are assigned to the dog class, which exhibited the highest level of confusion with the cat class in the original model. This aligns with the illustration shown in Figure 1 where the forget space gets redistributed to the classes in the proximity of the forget class. In the confusion matrix of the retrained model shown in Figure 8(c), we similarly observe a high number of cat samples being assigned to the dog class. Further, the confusion matrix for the our unlearned model revealed a reduction in correctly classified dog samples, the only class exhibiting this effect. This suggests that unlearning the cat class might have unintentionally removed features crucial for classifiying dogs, possibly due to their high inter-class confusion. We evaluate this behaviour further by studying the models trained on ImageNet.

**Table 3:** Effect of unlearning on the confusing classes for ImageNet dataset. See the results on Variant of ViT in Table 14

| Model | Original Accuracy | $ACC_r(\uparrow)$ | $ACC_f(\downarrow)$ | $MIA(\uparrow)$ | Top-5 Confusing Classes | |
|---|---|---|---|---|---|---|
| | | | | | Before - Unlearning | After - Unlearning |
| VGG11_BN | 68.61 | 65.287 | 0 | 100 | 63.6 | 14.4 |
| ViT_B_16 | 80.01 | 78.92 | 0 | 100 | 80.8 | 54.4 |

**Table 4:** Effect of Incomplete Retain Set for unlearning the Cat class from VGG11_BN model.

| Missing Class | $ACC_r(\uparrow)$ | $ACC_f(\downarrow)$ | $Drop\_Missing$ | % Confusion |
|---|---|---|---|---|
| Class 1 (airplane) | 93.88 | 0.0 | -0.1 | 0.8 |
| Class 2 (automobile) | 93.71 | 0.0 | +0.3 | 0.2 |
| Class 3 (bird) | 93.72 | 0.0 | +0.2 | 3.1 |
| Class 5 (deer) | 93.74 | 0.0 | -0.4 | 1.9 |
| Class 6 (dog) | 92.36 | 0.0 | -13.7 | 8.0 |
| Class 7 (frog) | 93.77 | 0.0 | -1.0 | 2.0 |
| Class 8 (horse) | 93.82 | 0.0 | -0.2 | 0.8 |
| Class 9 (ship) | 93.79 | 0.0 | -1.7 | 0.3 |
| Class 10 (truck) | 93.91 | 0.0 | -0.6 | 0.6 |

**Inter-Class dependency:** To study the Inter-Class dependency, we focused on the "Tibetan Terrier" class (class 200) within the ImageNet dataset, which contains 118 dog breeds (classes 151 "Chihuahua" to 268 "Mexican Hairless"). Using the confusion matrix, we identified the top 5 classes most frequently confused with "Tibetan Terrier." We then tracked the accuracy of these classes before and after unlearning "Tibetan Terrier." Table 3 summarizes the results. In our experiment, we defined "confusion" with a class as the percentage of images from the "Tibetan Terrier" class that were misclassified as the target class. Notably, "Shih Tzu" and "Lhasa Apso" exhibited the highest confusion with "Tibetan Terrier," at around $\sim 10\%$ and $\sim 8\%$, respectively for the ViT_B_16 model. Consistent with our earlier observations, unlearning the "Tibetan Terrier" class led to a decrease in performance for classes that were frequently confused with it ( or is hierarchically closer to).

**Effect of Incomplete Retain Set:** In Table 4, we analyze the effect of incomplete retain set, particularly, our algorithm does not have access to one of the classes in the retain set, this is referred to as "Missing Class". We report the test accuracy of the retain classes ($ACC\_r$) which includes the missing class, the accuracy of unlearned class($ACC\_f$), the drop in accuracy of the miss class ($Drop\_Missing$), and the confusion between the missing class and the unlearned class (obtained from Figure 8). The results show that there is a maximum drop in accuracy when the dog class is missing from the retain set. As it is maximally confused

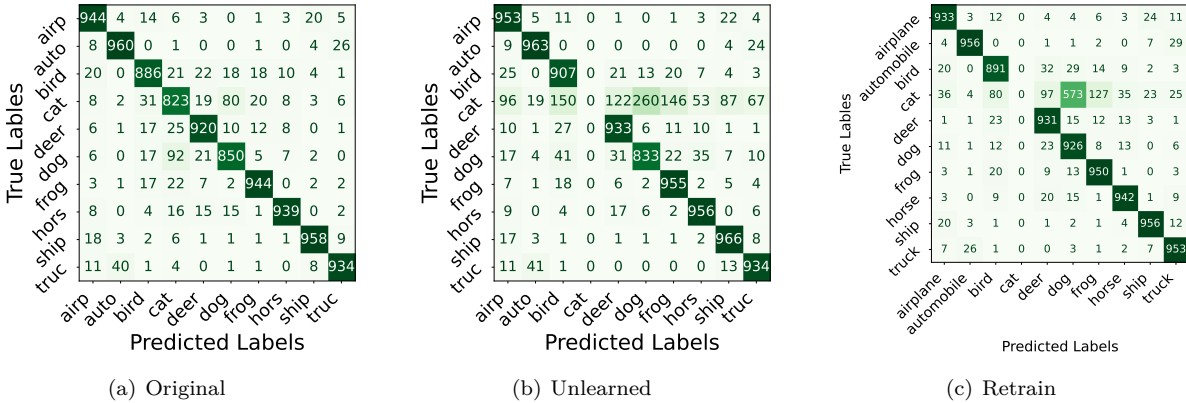

(a) Original       (b) Unlearned       (c) Retrain

**Figure 8:** Confusion Matrix for original VGG11 model and model unlearning cat class, showing redistribution of cat samples to other classes in proportion to the confusion in original model.

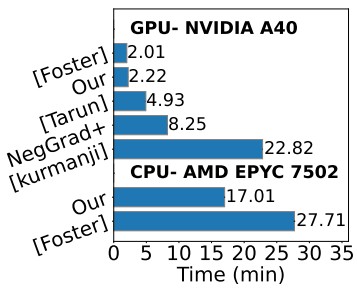

**Figure 9:** Runtime for different algorithms unlearning single class from ViT_B_16 on ImageNet dataset.

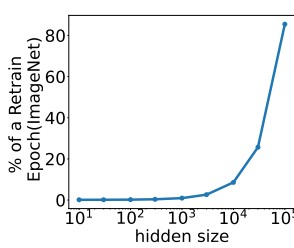

**Figure 10:** Change in compute with an increase in hidden size for a ViT_B_16 model on ImageNet dataset.

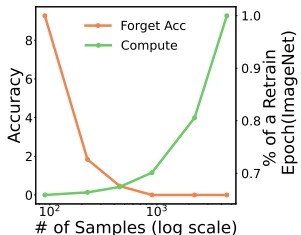

**Figure 11:** Change in Forget Accuracy and Compute with the number of retain and forget samples Used for SVD a ViT_B_16 model on ImageNet dataset.

with the target cat class. Removal of cat class without access to the dog information (or confusing class information) catastrophically affects the model usability on dog class.

**Runtime:** We evaluate the runtime performance of different unlearning algorithms. Figure 9 presents the average runtime over three trials for unlearning class 0 from a ViT model trained on the ImageNet dataset. We utilize implementations from the algorithms' official repositories and assess their runtime on a single Nvidia A40 GPU. Our algorithm achieves favorable runtime compared to most iterative methods considered in this work. Notably, the method referenced in Foster et al. (2024) exhibits the fastest runtime in this setting. However, the runtime advantange of Foster et al. (2024) diminishes significantly when compared on an AMD EPYC 7502 32-core CPU. We observe that Foster et al. (2024) is approximately 60% slower than our algorithm.

**Impact of increasing hidden size on Compute:** Since Singular Value Decomposition (SVD) is computationally expensive, increasing the hidden layer size of the network is likely to significantly impact the computational cost of our algorithm. We analytically calculate the computational cost for our algorithm (see Appendix A.13) to investigate this relationship in Figure 10. The figure plots the normalized computational cost (relative to 1 epoch of retraining) of our approach on the y-axis, for different hidden layer sizes shown on the x-axis. As expected, the figure demonstrates a positive correlation between the computational cost and the hidden layer size. Impressively, even for a hidden layer size of $100,000$, our method's computational cost remains lower than that of a single retraining epoch. In practice, for the widest know vision transformer model, ViT_G Zhai et al. (2022), with a hidden size of 1280, our method achieves a computational cost of 1.17% of a single retraining epoch.

**Number of Samples analysis:** In Figure 11 we plot the change in in model performance with sample set size. The sample set size for Retain Set and the Forget Set is kept the same in this experiment. We vary the number of samples as [90, 225, 450, 900, 2250, 4500] and present the accuracy on the forget set and the compute cost on . We observe that we need around 900 samples of retain and forget set each to obtain $ACC_f = 0$. This translates to around 100 samples pre retain class which is similar to what is used in Saha et al. (2021).

**One-Shot Multi Class Forgetting:** The objective of Multiclass removal is to remove more than one class from the trained model. In multi task learning a deep learning model is trained to do multiple tasks where each of the tasks is a group of classes. The scenario of One-Shot Multi-Class where the unlearning algorithm is expected to remove multiple classes in a single unlearning step has a practical use case in such task unlearning. Our algorithm estimates the Retain Space $U_r$ and the Forget Space $U_f$ based on the samples from $X_r$ and $X_f$. It is straightforward to scale our approach to such a scenario by simply changing the retain sample $X_r$ and $X_f$ to represent the samples from class to be retained and forgotten respectively. We demonstrate multi class unlearning on removing 5 classes belonging to a superclass on CIFAR100 dataset in Figure 12. We observe our method is able to retain good accuracy on Retain samples and has above 95% MIA accuracy while maintaining a low accuracy on forget set under this scenario.

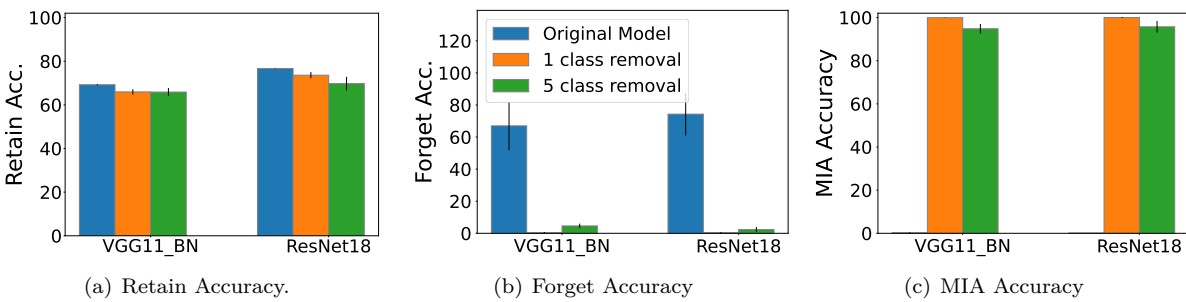

(a) Retain Accuracy.  (b) Forget Accuracy  (c) MIA Accuracy

**Figure 12:** One-shot Multi-Class Unlearning for CIFAR100 dataset.

When compared with Tarun et al. (2023) under this unlearning setting (see Table 15 in Appendix A.14) we see our method has significantly better performance. We also present results of multiclass unlearning on CIFAR10 in Appendix A.14, which shows similar trends.

**Sequential Multi Class Forgetting:** This scenario of multiclass unlearning demonstrates another practical use case of our algorithm where different unlearning requests come at different instances of time. In our experiments, we sequentially unlearn classes 0 to 9 in order from the CIFAR10 dataset on VGG11 and ResNet 18 model. The retain accuracy of the unlearned model is plotted in Figure 13. The forget accuracy for all the classes in the unlearning steps was zero. We observe an increasing trend in the retain accuracy for both the VGG11 and ResNet18 models which is expected as the number of classes reduces or the classification task simplifies.

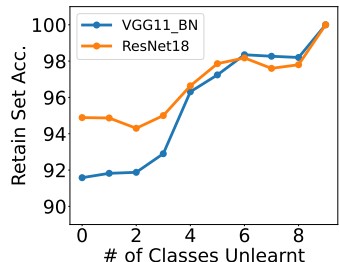

**Figure 13:** Sequential class removal on CIFAR10 dataset.

## 7 Conclusion

In this work, we present a novel class and multi-class unlearning algorithm based on Singular Value Decomposition (SVD). This approach eliminates the need for computationally expensive and potentially unstable gradient-based methods, which are prevalent in existing unlearning techniques. We demonstrate the effectiveness of our SVD-based method across various image classification datasets and network architectures. Our analysis using saliency-based explanations confirms that no class-discriminatory features are retained after unlearning. Additionally, confusion matrix analysis verifies the redistribution of unlearned samples based on their confusion with other classes.

Traditional unlearning methods rely heavily on gradients, leading to limitations in computational cost and sample size requirements, especially for large models and datasets. Our work addresses these limitations by introducing a novel and demonstrably superior solution for class-wise unlearning, outperforming state-of-the-art baselines. Our approach is radically different for many of the current approaches as our algorithm (a) is Gradient-Free, (b) uses Single-step weight update, (c) has novel weight update mechanism and (d) is compute and sample efficient. Our in-depth analyses, including investigations into layer importance, hyperparameters, scalability, and output distributions, were driven by the novelty and potential of our approach. We believe this work has the potential to revolutionize the field of efficient unlearning, inspiring future explorations across diverse domains and expanding the breadth of this research area.

## 8 Impact Statement

This research addresses crucial ethical aspects and societal consequences in the domain of machine learning, specifically focusing on the emerging need for unlearning algorithms in the context of data privacy. By using few samples for unlearning, the study not only pushes the boundaries of machine unlearning but also sticks

to ethical principles about privacy and user rights. The proposed algorithm changes how we unlearn data from a learned model, making it scalable, especially for huge datasets like ImageNet. This energy-efficient approach marks a substantial step in minimizing overall environmental impacts, representing progress towards more sustainable machine learning practices. In essence, this work guides us toward responsible and privacy-conscious machine learning, making sure our futuristic technology respects both societal values and regulations.

## Acknowledgement

This work was supported in part by, the Center for Brain-inspired Computing (C-BRIC), the Center for the Co-Design of Cognitive Systems (COCOSYS), a DARPA-sponsored JUMP center, the Semiconductor Research Corporation (SRC), the National Science Foundation, and DARPA ShELL.

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

# A   Appendix

## A.1   Demonstration with Toy Example

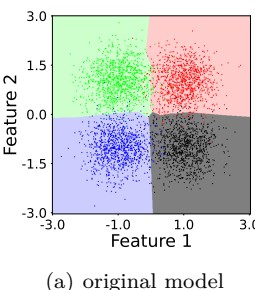 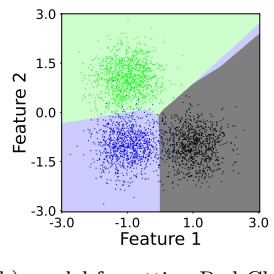 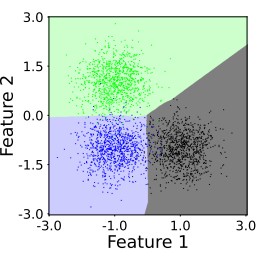

(a) original model              (b)  model forgetting Red Class              (c)  Retrained model

**Figure 14:** Illustration of the unlearning algorithm on a simple 4 class classification problem. Figure shows the decision boundary for (a) original model, (b) our unlearned model redistributing the space to nearby classes and (c) Retrained model without red class. Note - This figure is same as Figure 1 and duplicated here for ease of reference.

In Figure 14 we demonstrate our unlearning algorithm on a 4 way classification problem, where the original model is trained to detect samples from 4 different 2-dimensional Gaussian's centered around (1,1), (-1,1), (-1, -1) and (1, -1) respectively with a standard of (0.5,0.5). The training dataset has 10000 samples per class and the test dataset has 1000 samples per class. The test data is shown with dark points in the decision boundaries in Figure 14. We use a simple 5-layer linear model with ReLU activation functions. All the intermediate layers have 5 neurons and each layer excluding the final layer is followed by BatchNorm. We train this network with stochastic gradient descent for 10 epochs with a learning rate of 0.1 and Nestrove momentum of 0.9. The decision boundary learned by the original model trained on complete data is shown in Figure 14(a) and the accuracy of this model on test data is 95.60%. In Figure 14(b) we plot the decision boundary for the model obtained by unlearning the class with mean (1,1) with our algorithm. This decision boundary is observed to be close to the decision boundary of the model retrained without the data points from class with mean (1,1) as shown in Figure 14(c). The accuracy of the unlearned model is 97.43% and retrained model is 97.33%. This illustration shows that the proposed algorithm redistributes the input space of the class to be unlearned to the closest classes.

## A.2   Parameters in different layers

The majority of trainable parameters in CNNs (e.g., VGG, ResNets) and the key components of transformer architectures like ViTs reside within linear and convolutional layers (see Table 5. We focus on these layers for unlearning as they likely hold the most significant memorized information. *Other model parameters such as the normalization layers and the positional embeddings (in ViTs) are not updated in our algorithm.

## A.3   Training Details for CIFAR10/100 Models

All CIFAR10 and CIFAR100 models are trained for 350 epochs using Stochastic Gradient Descent (SGD) with the learning rate of 0.01. We use Nesterov (Sutskever et al., 2013) accelerated momentum with a value of 0.9 and the weight decay is set to 5e-4.

| Type of Layer | VGG11__BN (CIFAR10) | ResNet18(CIFAR10) | ViT__B__16(ImageNet1k) |
|---|---|---|---|
| Conv/Linear | 9750922 (99.94%) | 11164362 (99.91%) | 86377192 (99.78%) |
| Others | 5504 (0.06%) | 9600 (0.09%) | 190464 (0.22%) |
| Total Parameters | 9756426 | 11173962 | 86567656 |

**Table 5:** Fraction of parameters in Linear and Convolutional layers.

---

**Algorithm 2** NegGrad Algorithm

---

**Input:** $\theta$ is the parameters of the original model, $\mathcal{L}$ is the loss function, $\mathcal{D}_{train\_sub\_f}$ is the subset of the forget partition of the train dataset; and $\eta$ is the learning rate

1. procedure **Unlearn**( $\theta$, $\mathcal{L}$, $\mathcal{D}_{train\_sub\_f}$, $\eta$ )
2.     $\theta_f^* = \theta$
3.     **for** step = 1,...., 500 **do**
4.         input, target = **get_batch**($\mathcal{D}_{train\_sub\_f}$)
5.         $g$ = **get_gradients**($\theta_f^*$, $\mathcal{L}$, input, target)
6.         $g$ = **gradient_clip**($g$,1.0)
7.         $\theta_f^* = \theta_f^* + \eta g$
8.         **if** step multiple of 100 **do**
9.           $acc_f$ = **get_accuracy**($\theta$, $\mathcal{D}_{train\_sub\_f}$)
10.         **breakif** $acc_f < 0.1$
11. **return** $\theta_f^*$

---

**Algorithm 3** NegGrad+ Algorithm

---

**Input:** $\theta$ is the parameters of the original model; $\mathcal{L}$ is the loss function; $\mathcal{D}_{train\_sub\_f}$ and $\mathcal{D}_{train\_sub\_f}$ are the subset of the retain and forget partition of the train dataset respectively; and $\eta$ is the learning rate.

1. procedure **Unlearn**( $\theta$, $\mathcal{L}$, $\mathcal{D}_{train\_sub\_r}$, $\mathcal{D}_{train\_sub\_f}$, $\eta$ )
2.     $acc_f$ = **get_accuracy**($\theta$, $\mathcal{D}_{train\_sub\_f}$);     $\theta_f^* = \theta$
3.     **for** step = 1,...., 500 **do**
4.         **if** $acc_f > 0.1$ **do**
5.           input, target = **get_batch**($\mathcal{D}_{train\_sub\_f}$)
6.           $g_a$ = **get_gradients**($\theta_f^*$, $\mathcal{L}$, input, target)
7.           $g_a$ = **gradient_clip**($g_a$,1.0)
8.         **else**
9.           $g_a = 0$
10.         input, target = **get_batch**($\mathcal{D}_{train\_sub\_r}$)
11.         $g_d$ = **get_gradients**($\theta_f^* f$, $\mathcal{L}$, input, target)
12.         $\theta_f^* = \theta_f^* + \eta g_a - \eta g_d$
13.         **if** step multiple of 100 **do**
14.           $acc_f$ = **get_accuracy**($\theta$, $\mathcal{D}_{train\_sub\_f}$)
15. **return** $\theta_f^*$

---

### A.4 NegGrad Algorithm

In the NegGrad approach, we finetune the model for a few steps using gradient ascent on the forget partition of the train set $D_{train\_f}$ with a gradient clipping threshold set at 1.0. Pseudocode for NegGrad is presented in Algorithm 2. The algorithm initialized the unlearn parameters $\theta_f^*$ to the original parameters $\theta$ and does 500 steps of gradient ascent on the forget subset of the training data. After every 100 steps, we evaluate the model accuracy on $\mathcal{D}_{train\_sub\_f}$ and exit ascent when $acc_f$ becomes lower than 0.1. This restricts the gradient ascent from catastrophically forgetting the samples in the retain partition.

### A.5 NegGrad+ Algorithm

This algorithm does gradient ascent on the forget samples and gradient descent on the retain samples for 500 steps. NegGrad+ is a superior gradient ascent unlearning algorithm as compared to the NegGrad. Algorithm 3 outlines the pseudocode for the NegGrad+ unlearning approach. The algorithm initializes the unlearn parameters $\theta_f^*$ to the original parameters $\theta$ and gets the model accuracy on the forget partition $acc_f$. The gradients $g_a$ are computed on the forget partition if the $acc_f$ is greater than 0.1 otherwise $g_a$ is set to 0. The gradient on the retain batch denoted by $g_d$ is computed at every step and the unlearn parameters

**Table 6:** Hyperparameters for our approach with single class unlearning or sequential multi-class unlearning.

| Dataset | alpha_r_list | alpha_f_list | samples/class in $X_r$ | samples/class class in $X_f$ |
|---|---|---|---|---|
| CIFAR10 | [10, 30, 100, 300, 1000] | [3] | 100 | 900 |
| CIFAR100 | [100, 300, 1000] | [3, 10, 30, 100] | 10 | 990 |
| ImageNet | [30, 100, 300, 1000, 3000] | [3, 10, 30, 100, 300] | 1 | 500 |

**Table 7:** Best hyper-parameters for our algorithm .

| Dataset | Architecture | alpha_r in $X_r$ | alpha_f in $X_f$ |
|---|---|---|---|
| CIFAR10 | VGG11_BN/ResNet18 | 100 | 3 |
| CIFAR100 | VGG11_BN/ResNet18 | 1000 | 30 |
| ImageNet | VGG11_BN | 3000 | 30 |
| ImageNet | ViT_B_16/ViT_l_16 | 100 | 10 |
| ImageNet | ViT_H_16/ViT_B_32 / ViT_L_32 | 300 | 30 |

**Table 8:** Hyperparameter tuning space for NegGrad, NegGrad+ and (Tarun et al., 2023) benchmarks.

| Method | $\eta$ or $\eta_{repair}$ or $\eta_{impair}$ |
|---|---|
| NegGrad | [1e-4, 2e-4,5e-4,1e-3,2e-3,5e-3,1e-2] |
| NegGrad+ | [1e-4, 2e-4,5e-4,1e-3,2e-3,5e-3,1e-2] |
| Tarun et al. (2023) | [1e-4, 2e-4,5e-4,1e-3,2e-3,5e-3,1e-2] |

**Table 9:** Hyperparameters ranges for Kurmanji et al. (2023).

| Dataset | Learning-rate | forget-set batch size | retain-set batch size |
|---|---|---|---|
| CIFAR10/CIFAR100/ImageNet | $[10^{-4}, 5 \times 10^{-4}, 10^{-3}]$ | [32, 64, 128, 256] | [32, 64, 128, 256] |

**Table 10:** Hyperparameters ranges for Foster et al. (2024).

| Dataset | $\lambda$ | $\alpha$ |
|---|---|---|
| CIFAR10/CIFAR100/ImageNet | [0.1,0.3,1,3,5] | [0.1,0.3,1,3,10,30,100] |

are updated in the descent direction for the retain samples and ascent direction for the forget samples. The values of $acc_f$ is updated after every 100 steps. This algorithm mitigates the adverse effect of Naive descent on the retain accuracy. Once the model achieves the forget accuracy less than 0.1 the algorithm tries to recover the retain accuracy by finetuning on the retain samples.

## A.6 Hyperparameter Discussion

Our approach introduces four key hyperparameters: the list of $\alpha_r$ values (alpha_r_list), the list of $\alpha_f$ values (alpha_f_list), and the number of samples used to estimate the Retain Space and Forget Space. The values for these hyperparameters are dependent on the dataset and are presented in Table 6 of Appendix A.6. Best hyperparameters for the results of Table 1, Table 2, and Table 11 are presented in Table 7 The NegGrad and NegGrad+ require tuning of the learning rate $\eta$ for atleast 1 unlearning class and a list of the learning rates is presented in Table 8 in Appendix A.6. We tune this hyperparameter for unlearning the first class on each model-dataset pair. Once determined, these hyperparameters remain fixed for unlearning all other classes. The SoTA (Tarun et al., 2023) baseline introduces 2 learning rates for the impair and the repair stages represented by $\eta_{impair}$ and $\eta_{repair}$. Similar to the other baselines these hyperparameters are only tuned on one class for each model-dataset pair. For (Kurmanji et al., 2023) we use all the suggested hyperparameters (not explicitly mentioned) given in the work for Large scale experiments on CIFAR10 for class unlearning-type (Table 3). We tune the batch sizes (forget set bs and retain set bs) and learning rate as given in Table 9. We restrict the number of minimization steps to 3 and maximization steps to 2, in order to align the compute

budget with that of our algorithm. We were unable to train a satisfactory model with Kurmanji et al. (2023) algorithm for ViT under the above constaints and hence had to increase the minimization steps to 10 and maximization steps to 5 (only of $ViT\_B\_16$). Lastly, for Foster et al. (2024) we tuned the hyperparameter $\lambda$ in ranges given in Table 10 for all the experiments.

The Retraining method does not add any additional hyperparameters and is trained with the same hyperparameters as the original model.

### A.7  MIA Attack Details

The goal of the MIA experiment was to demonstrate how the unlearned models behave as compared to the Retrained model and the original model. Below we mention the details of MIA experiments.

**Training** - We train a Support Vector Machine (SVM) classifier as a MIA model to distinguish between $D_{train\_r}$(as class 1 or member class) and $D_{test\_r}$(as class 0 or nonmember class).

**Testing** - We show this SVM model $D_{train\_f}$ to check if the MIA model classifies it as a member or nonmember. When the MIA model classifies it class 0 (Nonmember) then the MIA model believes that the samples from $D_{train\_f}$ do not belong to the Train set. This is what is meant by having a high accuracy on $D_{train\_f}$.

**Interpretations of MIA scores**- We use the training and testing procedures mentioned above for all the models. Below we present interpretation for different models

- Original model - We see that the original model has a low MIA score (nearly 0) which means the SVM model classifies $D_{train\_f}$ as member samples. This is expected as $D_{train\_f}$ belonged to the training samples.

- Retrained model - We see that the Retrained model has a high MIA score (100%) which means the SVM model classifies $D_{train\_f}$ as nonmembers. This is expected as $D_{train\_f}$ does not belong to the training samples.

- Unlearned model - By these experiments of MIA we wanted to see how MIA scores of unlearned models perform. We observe the model unlearned with our algorithm consistently performs close to the retrained model as compared to other baselines.

### A.8  Experiments on Variants of ViT

Table 11 shows the results for our algorithm for different versions of ViT model from the pytorch library. The experimental setup is identical to the one presented in Table 2.

**Table 11:** Single class forgetting with our algorithm on ImageNet-1k dataset for variants of ViT.

| Method | Original Accuracy | $ACC_r(\uparrow)$ | $ACC_f(\downarrow)$ | $MIA(\uparrow)$ |
|---|---|---|---|---|
| ViT_B_16 | 80.01 | $78.47 \pm 0.84$ | $0.2 \pm 0.63$ | $99.98 \pm 0.05$ |
| ViT_L_16 | 78.83 | $78.48 \pm .26$ | $0.22 \pm 0.67$ | $99.78 \pm 0.01$ |
| ViT_H_14 | 85.48 | $84.35 \pm 0.40$ | $0.2 \pm 0.63$ | $97.0 \pm 0.05$ |
| ViT_B_32 | 73.48 | $71.02 \pm 0.77$ | $0.4 \pm 0.84$ | $98.0 \pm 0.02$ |
| ViT_L_32 | 75.00 | $73.64 \pm 0.36$ | $0.2 \pm 0.63$ | $97.4 \pm 0.03$ |

### A.9  Variants of Our algorithm

This section presents two variants of the algorithm depending on the location of the activation suppression. Consider the linear layer $a_o = a_i \times \theta^T$, where $a_o$ and $a_i$ are the input activation and output activations of a linear layer. The algorithm presented in the main paper focuses on activations before the linear layer,

**Table 12:** Location of Projection. Experiments on CIFAR10 dataset similar to Table 1

| Method | VGG11__BN | | ResNet18 | |
|---|---|---|---|---|
| | $acc_r$ | $acc_f$ | $acc_r$ | $acc_f$ |
| Original | 91.58 | | 94.89 | |
| input activation suppression (main paper) | $91.77 \pm 0.69$ | 0 | $94.19 \pm 0.50$ | $0.03 \pm 0.09$ |
| output activation suppression | $90.73 \pm 1.28$ | $0.15 \pm 0.38$ | $91.44 \pm 1.22$ | $1.05 \pm 1.13$ |
| both | $91.51 \pm 0.68$ | 0 | $93.96 \pm 0.60$ | $0.21 \pm 0.45$ |

i.e. the input activations $a_i$. We could also suppress the output activations. This activation suppression meant projecting the parameters on the orthogonal discriminatory projection space $(I - P_{dis})$, which is post multiplying the parameters $\theta$ with $(I - P_{dis})^T$. Now if we were to suppress the output activations $a_o$ it would be the same as pre-multiplying the parameters $\theta$ with $(I - P_{dis})^T$. (Note, for suppressing $a_o$ the output activations are used to compute $P_{dis}$). This variant of our approach is capable of removing the information in the bias and normalization parameters of the network. The other variant suppresses both the input and output activations using their respective projection matrices. The results for these variants are presented in Table 12. We observe that the performance of these two variants is lower than the algorithm in the main paper and hence do not analyze it further.

### A.10 Explaining Trends in Hyperparameter Variation

Equation 2 in the paper helps us understand how $\alpha$ directly controls the importance of basis vectors. Increasing $\alpha$ amplifies their significance while decreasing alpha diminishes it. The overall update equation of our algorithm for layer weights $\theta$ can presented in Equation 4. We explain the conclusions of Figure 4 in this Subsection.

#### A.10.1 Trends in Figure 4(a) (varying $\alpha_r$)

**Retrain Accuracy:** If we assume $P_f$ is Identity when we are studying the effect of $\alpha_r$ in Equation 4 simplifies to $\theta_f = P_r\theta$. Now, decreasing $\alpha_r$ weakens( or scales down ) basis vectors in Ur. This translates to reduced retrain accuracy as $P_r$ shrinks towards zero, leading to catastrophic unlearning of the retain partition.

**Forget Accuracy:** Expanding Equation 4 we get, $\theta_f = (I - P_f + P_f P_r)\theta$. Increasing $\alpha_r$ amplifies all basis vectors, including those overlapping between $U_r$ and $U_f$ (due to $P_f P_r$). This effectively boosts forget accuracy for high enough $\alpha_r$.

#### A.10.2 Trends in Figure 4(b) (varying $\alpha_f$)

**Retrain Accuracy:** Expanding Equation 4 we get, $\theta_f = (I - P_f + P_f P_r)\theta$. A decrease in $\alpha_f$ decreases all the basis vectors in $U_f$, including the ones overlapping with $U_r$ (due to $P_f P_r$). This effectively reduces retain accuracy for high enough $\alpha_f$.

**Forget Accuracy:** If we assume that $P_r$ is a zero matrix for studying the effect of $\alpha_f$, Equation 4 would simplify to $\theta_f = (I - P_f)\theta$. Increasing $\alpha_f$ diminishes the importance of basis vectors within the forget dataset (due to $I - P_f$), resulting in the observed decrease in forget accuracy.

### A.11 Saliency Maps for CIFAR10 on VGG Dataset

Table 13 shows the Saliency maps for all the classes that are not presented in the main paper (Figure 7).

### A.12 Effect of Inter-Class Confusion for variants of ViT

Table 14 shows the drop in accuracy of top-5 confusing classes for unlearning "Tibetan Terrier" for the ViT model trained on ImageNet dataset.

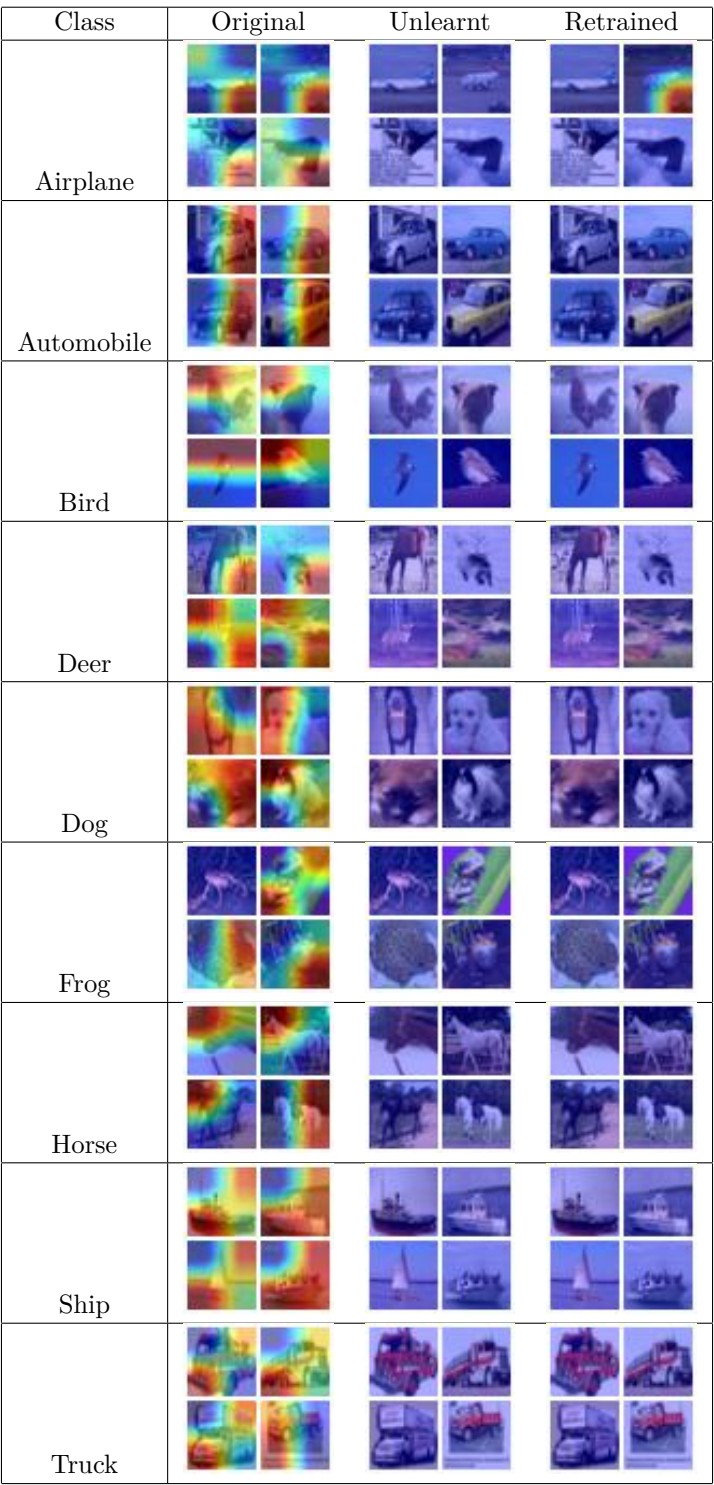

**Table 13:** Saliency map for CIFAR10 dataset on VGG11 Network. Similar to Figure 7

**Table 14:** Effect of unlearning on the confusing classes for ImageNet dataset for variants of ViT.

| Model | Original Accuracy | $ACC_r(\uparrow)$ | $ACC_f(\downarrow)$ | $MIA(\uparrow)$ | Top-5 Confusing Classes | |
|---|---|---|---|---|---|---|
| | | | | | Before - Unlearning | After - Unlearning |
| ViT_B_16 | 80.01 | 78.92 | 0 | 100 | 80.8 | 54.4 |
| ViT_L_16 | 78.83 | 78.61 | 0 | 100 | 79.6 | 68.0 |
| ViT_H_14 | 85.48 | 83.91 | 0 | 100 | 69.6 | 25.6 |
| ViT_B_32 | 73.48 | 70.25 | 0 | 100 | 67.6 | 30.4 |
| ViT_L_32 | 75.00 | 73.47 | 0 | 98.0 | 79.2 | 54.0 |

### A.13 Compute Analysis for Single Layer of ViT on ImageNet dataset.

#### A.13.1 Linear Layer Compute Equations

Here we analyze the compute required for a linear layer. Say we have a linear layer of size $f_{in} \times f_{out}$, where $f_{in}$ is the input features and $f_{out}$ is the output features. Let the retain set have $n_r$ samples. The input activation for this layer will hence be of size $n_r \times f_{in}$. Below we analyze the compute required by various algorithms in this setting. We substitute all the parameters in the equation to obtain the compute in terms of $f_{in}$ and $f_{out}$.

**Retraining:** The compute required for this method will be $n_r f_{in} f_{out}$ for forward pass and $2n_r f_{in} f_{out}$ for backward resulting in total compute given by Equation 5. For the ImageNet experiments, $n_r$ is approximately 128000. Note this is the compute for the single epoch.

$$C_{retrain}^{\text{Linear}}(f_{in}, f_{out}) = 3n_r f_{in} f_{out} = 3840000 f_{in} f_{out} \tag{5}$$

**Analytical compute for our algorithm:** We have access to $n_{our\_r}$ and $n_{our\_f}$ retain and forget samples in our algorithm and for ImageNet these are 999 and 500 respectively. Our approach can be broken into 4 compute steps, namely representation collection, SVD, Space Estimation, and weight projection. Representation collection requires forward pass on a few samples and can be compute cost can be computed as mentioned before. For a matrix of size $m \times n$ SVD has a compute of $mn^2$ Cline & Dhillon (2006) where $m > n$. Space Estimation and weight projection steps involve matrix multiplication. For a matrix A of size $m \times n$ and matrix B of size $n \times p$ the compute costs of matrix multiplication $A \times B$ is $mnp$.

$$C_{our}^{\text{Linear}}(f_{in}, f_{out}) = \underbrace{(n_{our\_r} + n_{our\_f})f_{in}f_{out}}_{\text{Representation Matrix}} + \underbrace{(n_{our\_r} + n_{our\_f})f_{in}^2}_{\text{SVD}} + \underbrace{2f_{in}^3}_{P_{dis}\text{ Computation}}$$
$$+ \underbrace{f_{in}^2 f_{out}}_{\text{Weight Projection}} \tag{6}$$
$$= 1499 f_{in} f_{out} + 1499 f_{in}^2 + 2f_{in}^3 + f_{in}^2 f_{out}$$

#### A.13.2 Compute for a layer of ViT

A layer of ViT$_{\text{Base}}$ has 4 layers of size 768, namely Key weights, Query weights, value weights, and output weights in the Attention layer. The MLP layer consists of a layer of size $768 \times 3072$ and $3072 \times 768$. The total compute for a layer of Vit would be given by Equation 7. Note this is a simplified model and ignores the compute of the attention and normalization layers. Adding compute for the attention mechanism would only benefit our method as we only compute this for representation collection, whereas the baseline methods would have this computation at every forward and backward pass. These equations are used to obtain the numbers for each of the methods in Figure 10 and Figure 11.

$$C^{\text{ViT\_Layer}} = 4C^{\text{Linear}}(768, 768) + C^{\text{Linear}}(768, 3072) + C^{\text{Linear}}(3072, 768) \tag{7}$$

### A.14 Multi class Unlearning

We run experiments for this scenario on the CIFAR10 dataset with VGG11 and ResNet18 models. Figure 15 presents the mean and standard deviations for retain accuracy and the forget accuracy for 5 runs on each

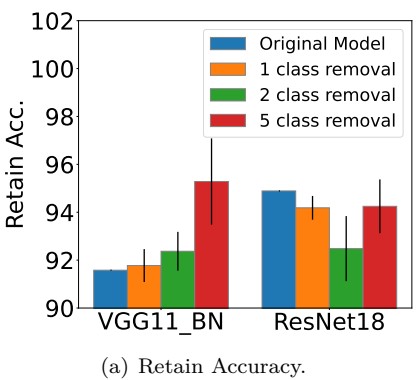

(a) Retain Accuracy.

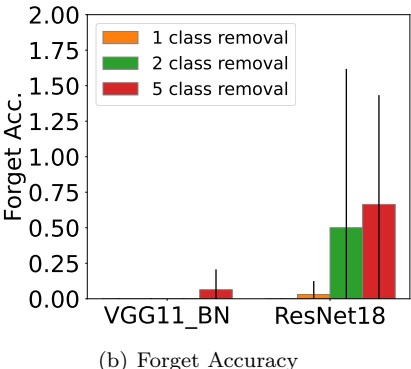

(b) Forget Accuracy

**Figure 15:** One-Shot Multi-Class Unlearning for CIFAR10 dataset.

configuration. The set of classes to be removed is randomly selected for each of these 5 runs. These results show our algorithm scales to this scenario without losing efficacy. Additionally we present the comparisons with baselines on the CIFAR100 dataset for unlearning a superclass in Table 15.

**Table 15:** Results for Multi class Forgetting on CIFAR100 dataset.

| Method | VGG11_BN | | | ResNet18 | | |
|---|---|---|---|---|---|---|
| | $ACC_r(\uparrow)$ | $ACC_f(\downarrow)$ | $MIA(\uparrow)$ | $ACC_r(\uparrow)$ | $ACC_f(\downarrow)$ | $MIA(\uparrow)$ |
| NegGrad+ | $57.75 \pm 1.40$ | $0.2 \pm 0.2$ | $0$ | $70.55 \pm 1.045$ | $0.7 \pm 0.12$ | $99.86 \pm 0.11$ |
| Tarun et al. (2023) | $54.31 \pm 1.09$ | $0$ | $0$ | $64.16 \pm 1.18$ | $1.34 \pm 2.33$ | $60.2 \pm 52.14$ |
| Ours | $65.94 \pm 1.89$ | $4.6 \pm 1.44$ | $94.8 \pm 2.2$ | $69.74 \pm 3.12$ | $2.33 \pm 1.61$ | $95.7 \pm 2.67$ |

