# OpenReview forum: "Deep Unlearning: Fast and Efficient Gradient-free Class Forgetting"
_TMLR — Accepted by TMLR_

### Review · Reviewer_6825 · 2024-05-29

**Summary Of Contributions:**

The paper studies the class machine unlearning problem where the target is to change a trained model's weights so that the model cannot make correct predictions for a certain forget class while retaining the performance on the rest of the classes. The paper proposes to use an SVD-based approach to modify the linear weights of a network by projecting them into a subspace so that the forget class will be unlearned. The paper shows good performance on image classification tasks on CIFAR and imagenet datasets.

**Audience:**

Yes

**Claims And Evidence:**

Yes

**Requested Changes:**

Minor:
1. Page 4 bottom: " Post multiplying the parameters with (I − Pdis)", missing l for Pdiss
2. Algorithm 1 input: "are is list"

**Strengths And Weaknesses:**

Strengths:
1. The paper proposes a method that is light-weighted to some extent and works well on image classification tasks.

Weaknesses:
1. I don't agree with the setting of the problem, even though there have been some papers published in such a field. Basically, for the class unlearning problem, if we want to completely forget a certain class, we can just change the weights of the forget class in the final classification layer (e.g., set the bias term to be a large negative number) or just simply hard-code that if the network's prediction is the forget class, we will output nothing (or any random class). Even though the method and its behaviors in the paper are interesting, given that there is such a simple solution, the good performance, as well as the purpose of studying this problem, are not that meaningful.

2. I don't agree with the point that the method does not require training. The tuning of the alpha list hyperparameters is essentially a form of training.

3. For the convolutional layer, I think the resulting equivalent linear matrix is typically very large (and also very sparse). I might not understand this point completely, but it seems to me that the projection matrix for convolutional layers could be very large, and using it as a matrix multiplication also makes the computation a lot slower than using the convolution operation.

---

> ### Author Response · Authors · 2024-06-23
> **Author's Response to Reviewer 6825's feedback**
>
> We thank the reviewer for their time and valuable feedback. We have updated the paper in light of the reviewer's suggestions and uploaded the revised manuscript with the changes highlighted in blue. Below we address the weaknesses raised by the reviewer.
> ## Weakness
> 1. **Problem Setup**
>     - **Author's Response**- To comply with regulatory demands like GDPR/CCPA, organizations must delete user data upon request. The motivation behind unlearning is to ensure this sensitive data is genuinely removed from the network's weights. The reviewer's suggested algorithms for class unlearning involve modifying the network independently of the forget data, making them easily reversible to retrieve the original model without access to the forget dataset. This approach deceives users into believing their data has been erased, while it can be recovered without their consent for reusing their data, violating the fundamental motivation of unlearning. In contrast, the algorithm's studied in this work modify the weights of all the layers using the retain and the forget data, requiring access to the forget data for relearning the information.  This ensures that the unlearning process is irreversible without the original data, aligning with ethical standards and regulatory requirements.
>
> 2. **Incorrect Claim of approach being Training-Free**
>     - **Author's Response**- We agree with the reviewer that the term "Training-Free" is misleading, as the process does involve tuning hyperparameters. We replaced "Training-Free" with "Gradient-Free" to more accurately reflect our method. Additionally, our algorithm relies on Singular Value Decomposition (SVD) to obtain the class discriminatory space, which can be considered a form of training, albeit consisting of a single update. We appreciate the reviewer's feedback and will make these adjustments to clarify our methodology.
>
> 3. **Clarification on our algorithm's implementation of Convolutional layer**
>     - **Author's Response**- We believe the reviewer may have misunderstood our approach to the representation matrix for convolution (presented in Section 4.1.1). It seems the reviewer thinks we use a Toeplitz matrix to convert convolution into a matrix-vector multiplication, resulting in a large and sparse matrix. While this formulation would work with our algorithm, it would indeed be highly inefficient, as the reviewer points out.
>
>         Instead, we use the unfold operation (natively supported in PyTorch - [``torch.nn.functional.unfold](https://pytorch.org/docs/stable/generated/torch.nn.functional.unfold.html)) on the inputs to convert the convolution into a matrix-matrix operation. This method aligns with the efficient implementation of convolution on Nvidia GPUs via the cuDNN library, which also converts convolutions into matrix-matrix multiplications for optimized performance on GPUs. Please refer to Figure 1 of [1] for further details.
>
> ### Minor
> We have addressed several typos in our work, including the one mentioned by the reviewer.
>
> ## Reference
> [1] Chetlur, Sharan, et al. "cudnn: Efficient primitives for deep learning." arXiv preprint arXiv:1410.0759 (2014).

---

> > ### Comment · Reviewer_d7Sc · 2024-06-27
> > **Weak evidence to support proposed unlearning process is irreversible, proposed method may suffer from the same issues**
> >
> > I believe the reviewer's critique is more a critique of the evaluations than the proposed method. Particularly, the evaluations show no evidence that the proposed method doesn't suffer from the problems mentioned in the response.
> >
> > 1. "The reviewer's suggested algorithms for class unlearning involve modifying the network independently of the forget data, making them easily reversible to retrieve the original model without access to the forget dataset."
> >
> > There's no evidence to show the proposed unlearning method will not get reversed if the model is trained further on new (not forget dataset) samples from the class.
> >
> > 2. " This approach deceives users into believing their data has been erased, while it can be recovered without their consent for reusing their data, violating the fundamental motivation of unlearning"
> >
> > The same could be true for the proposed method as we have no guarantees.
> >
> > Modifying weights in all layers through a process that takes as input the forget set does not guarantee anything about irreversibility and proper removal. Strong evaluations are needed to test these properties, which are not provided in the paper. The issue is that the trivial baseline proposed by the reviewer can score perfectly on the evaluations shown in the paper.

---

> > > ### Author Response · Authors · 2024-07-01
> > >
> > > We agree with the reviewer's concerns and have acknowledged this as a limitation in the updated manuscript (Section 4.3) as reiterated here:
> > >
> > > "Additionally, machine unlearning is an evolving field of research, and current evaluation techniques could be tricked by naive baselines, which may be insufficient to guarantee privacy Hayes et al. (2024). We hope future works address these shortcomings in evaluating the privacy provided by unlearning algorithms."

---

### Review · Reviewer_d7Sc · 2024-05-31

**Summary Of Contributions:**

The paper studies the setting of class-level unlearning using only a fraction of deleted/retain class samples, claiming this constraint allows more computational efficiency. They claim to propose the first retraining-free unlearning method, using SVD on the hidden layer activations, which they claim is state-of-the-art. Results are reported for VGG11, and ResNet on CIFAR10, CIFAR100 and VGG11, ViT on ImageNet. The method leads to good scores on their metrics for forgetting and maintaining utility. The paper also analyzes: the effect of hyperparameters, layer-level effects, gradcam visualizations, confusion matrices, and sequential, one-shot multi-class removal.

**Audience:**

Yes

**Broader Impact Concerns:**

None.

**Claims And Evidence:**

No

**Requested Changes:**

1. **Baselines**: Please add stronger baselines mentioned in the Weaknesses Section: SSD, EU-1, CF-1, Scrub with hyperparameter tuning
2. **Evaluations**: Please report a stronger MIA, and not just output metrics that can be tricked without parameter-level unlearning as highlighted in Weaknesses. Please add empirical unlearning times instead of analytical.
3. **Motivation**: Please motivate why unlearning with a few samples is important better.
4. **Related Work**: The paper misses a lot of prior work, some of which is highlighted in the Weaknesses, please include it in the discussion.
5. **Claims in Writing**: The writing currently makes claims bigger than the ones supported by the evidence, some of which is pointed out in the Weaknesses section, please fix this.

**Minor Issues**

“Three of these approaches…are the baselines used in the literature” - The citations here can be misleading. I guess you are trying to say the cited papers use the methods as baselines. But this is unconventional and thus confusing, as typically people cite the papers that proposed the methods/baselines.

The figure explaining MIA is hard to understand.

Figure 8-10 are underspecified. It’s not clear what dataset this analysis is done on.

Question: In one-shot multi-class removal, we saw a drop in retain accuracy. However it increases in sequential class removal. Why does this discrepancy exist, as the task should become easier even in the one-shot multi-class removal case.

**Strengths And Weaknesses:**

**Strengths**

1. Propose a novel SVD based unlearning method
2. Scale class-level unlearning experiments to ImageNet and Vision Transformers.
3. A commendable effort to analyze the effect of the method was made in Figures 3-12.
4. The paper is written clearly.

**Weaknesses**

1. **Constraint not motivated**: “Moreover, these unlearning algorithms may only have access to a fraction of the original training data, further complicating the unlearning process.”

The paper does not keep in mind the motivations for class-level unlearning, which leads to the addition of a seemingly arbitrary constraint of having access to only a fraction of original training data. Class-level unlearning is motivated by facial recognition/biometric systems, where a user is an entire class [1]. The user requests deletion of their entire data (the class), and its unclear why model developers would only have access to a fraction. Knowing partial deletion sets has been previously motivated for corrective unlearning [2], which also discusses why this is not really necessary for privacy oriented unlearning. The only provided motivation for this constraint is that using fewer samples leads to computational efficiency, but that’s an incorrect claim, as efficient algorithms can be designed while using the full forget set. See the next two points below.

2. **Overclaimed novelty**: “Our approach is radically different for the current approaches in the following ways:
a) Training-Free: We eliminate the computational burden and potential instability of gradient based unlearning.
b) Single-Step Weight Update: Our method achieves unlearning in a single update, surpassing the iterative nature of many baselines.”

This claim is incorrect. Training-free, single-step weight update unlearning methods have been proposed before, see SSD [3].

3. **Overclaimed efficiency**: “We achieve exceptional sample and compute efficiency as compared to SoTA approaches while improving the unlearning efficacy.”

Insufficient evidence is presented for this claim. First, the empirical unlearning times are not provided. Instead “analytical” times are provided in Figure 8, and it’s unclear how this analysis was done. For example, the proposed method’s time is taken as the same as 1 retain epoch. This seems incorrect, as apart from a forward pass, the method also requires SVD, an O(mn^2) operation. Overall, the efficiency claims would be less confusing if empirical unlearning times were provided instead. Further, please provide comparisons with SSD [3], which is a fast, retraining-free unlearing method that works with partial deletion sets. Please also provide comparisons with EU-1 and CF-1 that have shown 0 drop in Acc_r, and 0 Acc_f and are efficient as they only operate on a single layer [4]. Please also compare/cite few-shot unlearning by model inversion [5], though I guess that might not work well for large-scale datasets like ImageNet. Please also test how Scrub performs when a fraction of forget/retain set samples are known, and see how it works when the number of epochs is reduced (which would make it more efficient)

4. **Weak Evaluations**: “The unlearning algorithm should produce parameters that are functionally indistinguishable from those of a model retrained without the target class”

Empirically, the paper only looks at output-level evaluation metrics like accuracy on retain classes (Acc_r) and accuracy on forget classes (Acc_f), instead of evaluating whether the parameters are functionally indistinguishable. The MIA setup is not sufficient to indicate privacy protection. Please use sample-level MIA for evaluation, along with other recommendations in [6]. Why can’t output-level indistinguishability be achieved by a trivial baseline: Let D be the set of classes to be removed, in decoding the logits, ignore the class D probabilities. This would lead to no drop in Acc_r and 0 Acc_f, perfectly achieving the paper’s goals at 0 cost. This makes me believe that the evaluation setup in the paper is insufficient for measuring its goal, and the current setup allows for trivial baselines to achieve better results, making it hard to evaluate the efficacy of the method. Please add results for this baseline if you think this claim is not true. The main tables which compare prior methods are reported only on a single class. Since the improvements over existing methods are marginal, they may not be robust across different classes. Could you please report averaged results over 5-10 randomly sampled classes?

5. **Unfair comparisons wrt hyperparameters**:  “Further, our approach has fewer hyperparameters in comparison to the gradient-based baselines which are sensitive to the choices of optimizer, learning rate, batch size, learning rate scheduler, weight decay, etc.”

This statement is incorrect. For the gradient-based baselines, it is usually enough to use the same choices of the mentioned hyperparameters as the original training procedure. If anything, the method proposed requires 4 hyperparameters (alpha and no. of samples for both forget and retain set) which gradient-based approaches do not.

Scrub (Kurmanji et al.) shows better results than the proposed method in the ImageNet setting. In some other tables, it shows worse results, but I believe this is because hyperparameter tuning was not performed on Scrub (as is mentioned in the appendix, the suggested hyperparams in the original paper were taken). Please perform proper hyperparameter tuning on Scrub and report the best possible results that can be obtained with it, to have a fair comparison with the proposed method.

6. **Analysis claims not supported with sufficient evidence**: “We observe that the weight change is larger for the later layers. This suggests that the class discriminatory features are more prominent in the later layers of the network.”

This claim could be wrong, as its possible later layer weights just change more in the measured metric for all types of samples (instead of class discriminatory ones). A carefully designed control experiment would be needed to prove this claim.

**References**

[1] Machine Unlearning: Linear Filtration for Logit-based Classifiers, Springer Machine Learning 2022

[2] Corrective Machine Unlearning, ICLR DMLR Workshop 2024

[3] Fast Machine Unlearning without Retraining through Selective Synaptic Dampening, AAAI 2024

[4] Towards Adversarial Evaluations for Inexact Machine Unlearning, ArXiv 2023

[5] Few-Shot Unlearning by Model Inversion, ArXiv 2023

[6] Inexact Unlearning Needs More Careful Evaluations to Avoid a False Sense of Privacy, ArXiv 2024

---

> ### Author Response · Authors · 2024-06-23
> **Author's Response to Reviewer d7Sc's feedback - Part 1**
>
> We thank the reviewer for their time and valuable feedback. We have updated the paper in light of the reviewer's suggestions and uploaded the revised manuscript with the changes highlighted in blue. Below we address the changes requested by the reviewer.
>
> ## Requested Changes
> 1. **Baselines**
>     - **Author's Response** -  We have updated the paper as requested by the reviewer (See Table 1 and Table 2 for the results).  Below we address the specific points raised by the reviewer.
>         - Scrub Hyperparameter Tuning : The reviewer correctly points out that directly using the hyperparameters from the original paper leads to unsatisfactory class unlearning due to differences in datasets, architectures, and setups. We had to tune the forget-set and the retain-set batch sizes in order to get good unlearning performance. We have now updated the results in the paper with the following changes which led to better results for Scrub.
>
>             **a**. CIFAR - We now tune the learning rate in [1e−4,5e−4,1e−3] along with the batch sizes for the forget-set and the retain-set across [32,64,128,256].
>
>             **b**. ImageNet - Initially, we tuned the batch sizes for the retain-set and forget-set in the Scrub algorithm, using 2 maximization steps (ascent epochs) and 3 minimization steps (descent epochs) to align the compute budget with that of our algorithm. In this setting, we varied the learning rate across [1e−4,5e−4,1e−3,5e−3,1e−2] while varying the forget-set and retain-set batch sizes across [32,64,128,256] for ViT, but obtained unsatisfactory results.  We believe this is potentially due to an extremely low compute budget for an Iterative Gradient Descent/Ascent-based algorithm. A low number of steps forces the algorithm to have higher ascent steps to unlearn the forget-set, which is detrimental to the retain-set. Additionally, low maximization/minimization steps particularly affect ViTs as they have low inherent bias and require a larger iterations to converge. With this in mind, we have now increased the number of maximization steps to 5 and minimization steps to 10 for ViT, yielding improved results as shown in the table below. Note that this adjustment significantly increases the computational requirements. This can be observed in the runtime presented in the Table below (In response to reviewer's request for runtime).
>         - SSD Baseline Comparisons - We have included the results for the SSD algorithm on CIFAR and ImageNet in the Table 1 and Table2 of the paper. For these experiments, we tuned the hyperparameters Lambda in [0.1,0.3,1,3,5] and alpha in [0.1,0.3,1,3,10,30,100].
>
>             Similar to our algorithm, SSD performs gradient updates in a single step using Fisher information, rather than SVD on activations. The Fisher information is derived from the average weight gradients for each sample, where the weight gradients $\nabla W = X \times \delta$, with $X$ being the input activation and $\delta$ being the output gradient (or the output error signal).
>             A notable difference is that our algorithm uses only the input activations, while SSD scales the input activation with the output gradients ($\delta$). For a well-converged model, the loss surface is expected to be flat, resulting in output gradients carrying low information or being randomly oriented. This implies that SSD might require larger sample sizes to statistically compensate for this randomness. As observed in the Table 2, SSD performance drops significantly when the number of retain-set samples is reduced from 10,000 to 1,000. Note, this behavior is also observed by corrective unlearning [2] work. Additionally, we also observe that SSD algorithm has poor performance on VGG architecture. This could be due to the unreliable gradient information.
>
>         - EU-1 and CF-1 Comparisons- While we agree that EU-1 and CF-1 use the same hyperparameters as the training routine, making them more stable for unlearning, we observe that their training time and computational costs are at least 35% of retraining (CF-1 in Table 2 from the referenced paper [3]). Therefore, we believe these baselines fall into a different compute and latency regime than our algorithm. Additionally, our algorithm (or SSD[1]) can also be used as a better initialization for CF-k, potentially reducing the unlearning compute/time for their algorithm.

---

> > ### Comment · Reviewer_d7Sc · 2024-06-27
> > **Why have you not tuned alpha for Scrub?**
> >
> > I believe the most important hyperparameters for Scrub are actually alpha and the number of maximization (ascent) steps as defined in Kurmanji et al. I noticed these have not been tuned for the comparisons provided in this paper, which seems unfair. Could you please tune these and report best results?

---

> > > ### Author Response · Authors · 2024-07-01
> > > **Author's Response to Reviewer d7Sc's Comment on tuning $\alpha$**
> > >
> > > The reviewer correctly points out that the hyperparameters alpha ($\alpha$) might affect the unlearning performance. In Figure 5 of [1], we see the effect of $\alpha$ on the forget error (f-error) and retain error (r-error). However, our experiments with SCRUB showed the best performance for f-error (or $ACC_f$) and r-error (or $ACC_r$) with low variance across runs compared to all the algorithms in Table 1 and Table 2. Thus, we had not further tuned the hyperparameters $\alpha$ and $\gamma$.
> > >         Further, in the Table below we tune the hyperparameters $\alpha$ [0.001,0.003,0.01,0.03,0.1] for the CIFAR dataset on VGG11\_BN and  ResNet18 (keeping the other hyperparameters the same as Table 1 ) and did not observe a significant change in the results.
> > > - Table tuning hyperparameter $\alpha$ for CIFAR dataset.
> > >
> > >     |Dataset|Method||VGG11\_BN|||ResNet18||
> > >     |------- |------- | ------- | ------- | ------- |  ------- | ------- | ------- |
> > >     |||$ACC_r$|$ACC_f$|$MIA$|$ACC_r$|$ACC_f$|$MIA$|
> > >     |CIFAR10|Orignal |$92.09\pm0.89$ |0|0|$94.79\pm0.63$|0|0|
> > >     ||Tuning $\alpha$|$92.09\pm0.86$ |0|0|$95.05\pm0.58$|$1.03\pm2.21$|0|
> > >     || ||||||
> > >     |CIFAR100|Orignal |$69.09\pm0.3$ |0|0|$76.40\pm0.19$|0|0|
> > >     ||Tuning $\alpha$| $69.36\pm0.29$ |0|0|$76.52\pm0.22$|0|0|
> > >     || ||||||
> > >
> > >
> > > SCRUB has several hyperparameters that address the instability in the reliance on the gradient ascent algorithm. Some key hyperparameters that play an essential role in the SCRUB algorithm are learning rate, forget-set batch size, retain-set batch size, number of minimization steps, number of maximization steps, $\alpha$, $\gamma$,  weight decay, learning rate decay steps (or schedule), and the choice of optimizer. For our experiments, we tuned the learning rate ([1e-4, 5e-4, 1e-3]), forget-set batch size ([32, 64, 128, 256]), and retain-set batch size ([32, 64, 128, 256]). We observed that tuning these hyperparameters led to satisfactory performance across all our experiments in Table 1 and Table 2. To ensure a fair comparison, we limited the compute budget for the SCRUB algorithm by restricting the number of minimization and maximization steps, which resulted in good performance for SCRUB on the CIFAR dataset. Additionally, as suggested by the reviewer, we also tuned the minimization and maximization steps when we observed unsatisfactory unlearning performance (in the case of ViT). It is expensive to exhaustively tune every hyperparameter as SCRUB is computationally intensive with none of the compute being shared across the hyperparameters (especially for ImageNet dataset with ViT model where SCRUB take $\sim 30$ mins to unlearn).
> > >
> > > ### Reference
> > >  [1] Kurmanji, Meghdad, et al. "Towards unbounded machine unlearning." Advances in neural information processing systems 36 (2024).

---

> ### Author Response · Authors · 2024-06-23
> **Continuation of Author's Response to Reviewer d7Sc's feedback- Part 2**
>
> 2. **Evaluation**
>     - **Author's Response**
>         - **Stronger MIA** - We present the results for Stronger MIA as recommended in U-LIRA [4] in the updated version of our manuscript (see Figure 3 ).
>
>             We implemented U-LIRA [4] to evaluate our unlearning approach for CIFAR100 on a ResNet18 network and observe our method performs better than the baselines. To perform this evaluation, we trained eight models on the CIFAR100 dataset using ResNet18 as the *base model*, with the same hyperparameters as the original training. These models act as the base model for unlearning. We obtained the unlearnt model by applying the unlearning algorithms to these eight models. Next, we trained eight models without the forget-class in the training dataset, referred to as the *retrained models*.
>
>             Following the recommendations in U-LIRA [4], we applied various unlearning techniques to the eight base models to unlearn the target class giving us the *unlearnt model*. We then learnt the MIA model to distinguish between forget samples in the unlearnt model and the retained model (without the forget-set). Specifically, we used seven of these models to learn the distributions (mean and standard deviations) and used the eighth model pair to determine the detection threshold.
>
>             Finally, we tested the unlearnt model from Table 1 of our paper using the above U-LIRA [4] MIA attack. We used the forget set from the original model as specified in [4] and reported the accuracy of the MIA attack for identifying samples belonging to the forget-set. We conducted this analysis for all the target classes used in the paper (Table 1). Ideally, MIA accuracy should be close to 50%, indicating that the MIA attack fails to distinguish between the forget-set from the unlearnt and the retrained model (implying perfect class unlearning). The following table reports the mean and standard deviations across the 10 target classes for different baselines:
>
>             |Method| U-LIRA Accuracy |
>             |------- | ------- |
>             | Our   |$74.54 \pm 4.72$|
>             | Tarun et. al. |$76.34\pm3.63$|
>             | Scrub |$94.46\pm2.36$|
>             | SSD   |$94.76\pm2.73$|
>             | NegGrad+ |$96.34\pm2.5$|
>             | | | | | | | |
>     - **Runtimes**
>         We understand the importance of providing empirical unlearning times to substantiate our efficiency claims. In response to the reviewer's request, we have included the runtimes for different algorithms in Figure 9 of the updated manuscript. The runtimes are averaged over three runs for unlearning class 0 from a ViT model trained on the ImageNet dataset. It is important to note that runtimes are dependent on implementation and the hardware used. We used implementations of these algorithms from their official repositories and evaluate runtimes on Nvidia A40 GPU and AMD EPYC 7502 32-Core CPU.
>
>         - Runtime on a single Nvidia A40
>
>             |Method|Runtime(sec)|
>             |------- | ------- |
>             | SSD   |$120.80\pm0.74$|
>             | Our   |$133.15\pm0.60$|
>             | Tarun et al   |$295.66\pm1.59$|
>             | Scrub (low mix/max steps)|$419.38\pm1.01$|
>             | NegGrad+ | $494.75\pm0.65$|
>             | Scrub  |$1369.45\pm2.33$|
>             | | | | | | | |
>
>
>         - Runtime on AMD EPYC 7502 32-Core Processor
>
>            |Method|Runtime(sec)|
>            |------- | ------- |
>            | SSD   |$1662.68\pm7.08$|
>            | Our   |$1020.35\pm4.30$|
> 3. **Motivation**
>     - **Author's Response** - We have updated the motivation for unlearning with a few samples as requested by the reviewer. We reiterate this here - "In many practical scenarios, full dataset access is often restricted since companies typically only utilize the trained model in their applications. Additionally, data access is usually chargeable for each instance, making repeated access for unlearning purposes prohibitively expensive. Public datasets may also become unavailable for reuse in the future, and maintaining a copy of all training data is often costly and impractical. Furthermore, in the context of removing confusing or manipulated data from the model Goel et al. (2024); Schoepf et al. (2024), where the goal is to improve model generalization when the training dataset may contain misleading samples, obtaining the entire confusing/manipulated dataset is challenging and often compute-intensive. This limited access to data introduces additional challenges for the unlearning process. Consequently, several unlearning works explore sample-efficient approaches Golatkar et al. (2020a); Nguyen et al. (2020); Chundawat et al. (2023); Jeon et al. (2024)"

---

> ### Author Response · Authors · 2024-06-23
> **Continuation of Author's Response to Reviewer d7Sc's feedback- Part 3**
>
> 4. **Related Work**
>     - **Author's Response** - We have updated the related work section by adding the recent works SSD, EU-k, CF-k, Model Inversion as per reviewer's suggestion. We reiterate the changes here -  "Recent work on Exact Unlearning (EU-k) and Catastrophic Forgetting (CF-k) proposed in Goel et al. (2022) presents fine-tuning methods that update the last k layers of the network. EU-k re-initializes the last k layers and trains on the retain set, while CF-k directly fine-tunes the last k layers, leveraging learning dynamics for unlearning. Further, a few-shot unlearning approach proposed by Yoon et al. (2022) trains a generative model to create proxy data for the retain-set and forget-set. This algorithm unlearns by fine-tuning on the concatenated retain-set and relabeled forget-set. These algorithms are evaluated on small datasets and have high compute requirements, possibly restricting their use for large datasets. Selectively Synaptic Dampening (SSD) (Foster et al., 2024) introduces a method to unlearn using the Fisher information. Like our method, this approach enables unlearning in a single update step; however, it is shown to require a large dataset size Goel et al. (2024)."
>
> 5. **Claims in Writing**
>     - **Author's Response** - We have revised the following claims pointed out by the reviewer -
>         - **Novelty**
>
>             **a**. "Training-Free" - We update the claim of "Training-Free" to "Gradient-Free" in response to the reviewer 6825's Weakness point 2. While both SSD and our algorithm use the forget-set and retain-set data to achieve unlearning, neither can be considered entirely training-free. Our approach is distinct in that it relies on activations and does not use backpropagation-based weight gradients.
>
>             **b**.  "Single-Step Update"-We have revised the claim to: "Our method achieves unlearning in a single update, similar to Foster et al. (2024), and surpasses the iterative nature of many baselines."
>
>         - **Efficiency** - We have now provided the unlearning runtimes for different baselines in our experiments. Further, we observe that reducing the epochs and dataset size for gradient-based methods leads to a degradation in performance for ImageNet on a ViT model. Additionally, we would point out that hyper-parameter search for our algorithm reuses the majority of the compute (i.e. SVD) unlike many of the unlearning algorithms. We have modified the claim to "Our proposed algorithm is sample efficient while having low runtimes on large datasets like ImageNet."
>         - **Further, our approach has fewer hyperparameters in comparison to the gradient-based baselines which are sensitive to the choices of optimizer, learning rate, batch size, learning rate scheduler, weight decay, etc** - Any gradent-ascent based method would bring in instability and would be very sensitive to the hyper-parameters. This is due to the inherent catastrophic nature of gradient ascent which reduces the model performance on the retain-set if the unlearning algorithm is aggressive or does not completely unlearn on the forget-set when the unlearning algorithm is mild. This requires careful tuning of the hyperparameters and in some cases, it might not be possible to unlearn in a restricted compute budget. Additionally, we would like to point out we only tune $\alpha_r$ and $\alpha_f$ and do not explicitly tune the number of samples used for retain-set and forget-set for our approach. Further majority of the compute (i.e. SVD) for the hyper-parameter search is shared, unlike the gradient-based baselines. We have updated the statement to "Additionally, our approach has fewer hyperparameters to tune compared to gradient-ascent-based baselines, which are sensitive to choices of optimizer, learning rate, batch size, learning rate scheduler, weight decay, etc."
>
>         - **We observe that the weight change is larger for the later layers. This suggests that the class discriminatory features are more prominent in the later layers of the network.** - We agree with the reviewer, it is possible that all the unlearning primarily affects the later layers and further analysis would be required to verify this claim. Our claim is specific to Class Unlearning and might not be generally applicable to other scenarios of unlearning. Additionally, our statement about the weight change being larger in the later layers is consistent with existing literature that indicates initial layers learn more basic features, while later layers capture more complex, class-specific features [5]. We modify the above statement to "We observe that the weight change is larger for the later layers. This suggests that the class discriminatory features are more prominent in the later layers of the network which is consistent with the findings of Olah et al. (2017)."

---

> ### Author Response · Authors · 2024-06-23
> **Continuation of Author's Response to Reviewer d7Sc's feedback- Part 4**
>
> ## Minor Issues
> 1.  **Citations for Naive baselines** - We update the line to statement to clarify the confusion "Three of these approaches Retraining, NegGrad and NegGrad+ are common baselines used in literature such as (Tarun et al., 2023; Kurmanji et al., 2023). "
>
> 2. **MIA Figure** - We have updated the figure for readability.
> 3. **Figure 8-10 are underspecified.**- We add the dataset and the network in the caption.
> 4. **Question on drop in one-shot multi-class** - It is important to note that the two experiments are done on different datasets. One-shot multiclass unlearning presents results of CIFAR100 dataset while the sequential class removal was done on CIFAR10 dataset. We present one-shot multi-class removal for CIFAR10 in Figure 15 of appendix which shows increase in accuracy with removal.
>
> ## Reference
> [1] Fast Machine Unlearning without Retraining through Selective Synaptic Dampening, AAAI 2024.
>
> [2] Corrective Machine Unlearning, ICLR DMLR Workshop 2024.
>
> [3] Towards Adversarial Evaluations for Inexact Machine Unlearning, ArXiv 2023.
>
> [4] Inexact Unlearning Needs More Careful Evaluations to Avoid a False Sense of Privacy, ArXiv 2024
>
> [5] Chris Olah, Alexander Mordvintsev, and Ludwig Schubert. Feature visualization. Distill, 2017

---

> > ### Comment · Reviewer_d7Sc · 2024-06-27
> > **Commendable effort in alleviating concerns**
> >
> > I am impressed by the effort put in the response period to alleviate my initial concerns. The U-Lira MIA and computational efficiency results seem promising, and it is good to see comparisons with more baselines, and the claims adjusted accordingly. I think Weakness 2, 3 are resolved, and 4-6 are partially improved.
> >
> > I am still skeptical about the problem setup, motivation, and evaluations (Weakness 1, 4). As also highlighted by Reviewer 6825, trivial baselines can achieve good scores on the evaluations used, which makes me skeptical about progress in this class unlearning problem formulation. However, I understand that other papers studying this formulation suffer from the same issues, and hope this is addressed by future works.

---

> > > ### Author Response · Authors · 2024-07-01
> > > **Author's Response on Reviewer d7Sc's comment**
> > >
> > > We thank the reviewer for acknowledging our efforts and for raising these important concerns. We agree that the issues highlighted are valid and relevant to all the current unlearning baselines. Machine Unlearning is a relatively novel domain, and the problem setup, motivation, and evaluations are still evolving. Our work aims to contribute to this emerging field by providing a comprehensive comparison with more baselines, implementing stronger evaluation metrics like U-LIRA MIA, and adjusting claims to reflect our findings accurately. We acknowledge that trivial baselines can achieve good scores on the current evaluations, and we recognize the need for more robust and nuanced evaluation methods. We hope that future works will address these limitations and help refine the problem formulation and evaluation strategies in Machine Unlearning.
> > >
> > > Thank you for your feedback and for acknowledging the improvements made in our response. We have added a line acknowledging this limitation in our work:
> > >
> > >  "Additionally, machine unlearning is an evolving field of research, and current evaluation techniques could be tricked by naive baselines, which may be insufficient to guarantee privacy Hayes et al. (2024). We hope future works address these shortcomings in evaluating the privacy provided by unlearning algorithms."

---

### Review · Reviewer_gSKS · 2024-06-10

**Summary Of Contributions:**

Large pre-trained machine learning models are increasingly popular today. However, a lot of these models are capable of memorizing their training datasets, which creates safety and privacy related issues. Particularly, if any entity wants to remove their personal data from the training dataset of a model, merely removing it is not enough since the model might still remember the information from those data-points. On the other hand, computational costs are prohibitive enough that one cannot simply retrain these models from scratch on the left-over dataset. Hence, an increasingly important field of research is machine unlearning — where the goal is to make the model forget particular information without retraining and without destroying the model’s capacity on other tasks.

This paper studies machine unlearning and provides an effective algorithm for this task — the core contribution is to take a pretrained model and few examples for tasks the model needs to forget vs few examples for tasks the model needs to remember, uses SVD on layerwise activations to figure out features that are specific to the tasks that need to be forgotten, and use them to unlearn particular classes.

**Audience:**

Yes

**Broader Impact Concerns:**

No concerns.

**Claims And Evidence:**

Yes

**Requested Changes:**

The paper contains multiple typos that should be corrected. I would request the authors to answer my questions above, and also fix the following typos.

(Typo) Page 7, We can readily extended → we can readily extend

Our code available --> Our code is available

**Strengths And Weaknesses:**

# Strengths

1. The paper is nicely structured, with an interesting and novel method.
2. Performance across multiple datasets (CIFAR-10, CIFAR-100, ImageNet) and different architectures (VGG11, ResNet-18, ViT-B/16) is reported, this paper’s method works well in all settings. **Specifically, the paper mentions one setup with a Vision Transformer (though more examples here would be nice), where many other baselines fail, but this paper’s method works well**. This is a big strength of this paper in my opinion.
3. Nice set of ablations and analysis are presented.

# Weaknesses

I would list the following weaknesses/questions:

**(About figure 1)**

Figure 1 is pretty nice, illustrating the behavior of the algorithm. Is this just an illustration, or the actual effect of running the algorithm on a 4-class classification problem on a 2D space? If so, what underlying network is used for this task?

**(About preliminaries)**

> Mathematically, these parameters must satisfy $f(x_i, \theta^*) = f(x_i , \theta^*_f )$ for $(x_i , y_i) \in D_{test}$.

The above requirement for the class unlearning problem seems a bit too strict for me. Specifically, it makes sense that the two parameters behave the same on the retain set, i.e.,  $D_{test, r}$, but why do they need to behave the same on the forget set, $D_{test, f}$? As long as the accuracy on forget classes is low, I don’t see why $$f(x_i, \theta^*) \approx f(x_i , \theta^*_f )$$ has to hold for $(x_i , y_i) ∈ D_{test, f}$.  Also why use sampled datasets here for the problem definition and not distributions?

**(About Table 1)**

Why does the method Kurmanji et al. show such high standard deviations on CIFAR-10 for VGG11? Also why is there such a big performance difference between the two architectures on the retain classes?

I also don’t agree with the bolding: the bolding should be done according to the best number in each column, not according to some other metric that is not even presented in the main table directly. We understand each algorithm comes with a different tradeoff between accuracy on retain classes vs forget classes. Bolding each column separately makes it easier for the practitioners to choose their algorithms based on if they care about accuracy more for the retain classes vs forget classes. The retraining method can be put in a different color and mentioned that it is an upper bound.

**(About ViTs)**

One interesting other thing I notice: Tarun et al. and Kurmanji et al. does so much worse in terms of accuracy on forget classes for the ViTs. Is there anything specific about these algorithms that make them particularly bad for ViTs? This paper’s method seems to work with both ViTs and more traditional ResNets, which is a positive point for this paper. But more variants of ViTs/combination of more pre-trained variations would be great to establish this point.

**(Choice of alpha)**

Choosing $\alpha_f$ and $\alpha_r$ based on grid-search, despite working well, seems unsatisfactory. Could the authors report these hyper-param values on each (dataset, architecture) pair? How much do they vary?

**(More saliency-based analysis figures like Figure 6)**

Only a single qualitative analysis is given in Figure 6. Can at least a couple more be included in the Appendix?

**(Effect of having incomplete retain set)**

Imagine we want to forget 1 class, and remember the 9 other classes for CIFAR-10. However, the retain set only contains examples from 8 classes. How would this affect the performance of this paper’s method in terms of retain accuracy?

**(Inter-dependency of classes)**

Different classes of images might be related to each other (eg., different breed of dogs), and trying to unlearn one class might harm another class which is in the retain set. Do the authors have any qualitative/quantitative analysis on this? Datasets where class inter-dependency is easily defined (BREEDS [1]) can be an interesting place to test this.

# References

[1] BREEDS: Benchmarks for Subpopulation Shift, https://arxiv.org/abs/2008.04859

---

> ### Author Response · Authors · 2024-06-23
> **Author's Response to Reviewer gSKS's feedback - Part 1**
>
> We thank the reviewer for their time and valuable feedback. We have updated the paper in light of the reviewer's suggestions and uploaded the revised manuscript with the changes highlighted in blue. Below we address the weaknesses raised by the reviewer.
> ## Weakness/Questions
> 1. **About figure 1**  - We run our algorithm on the 4-class classification problem. We have presented the experimental details in Appendix A.1 and the code for this experiment in the supplementary (demo.py file).
>         Network Details - "We use a simple 5-layer linear model with ReLU activation functions. All the intermediate layers have 5 neurons and each layer excluding the final layer is followed by BatchNorm".
> 2.  **About preliminaries** - The goal of approximate unlearning algorithms is to ensure that the output distribution of an unlearnt model is indistinguishable from a model trained on the retain set. This implies that the unlearnt model should not leak any distinguishable information about the forget-set, which is crucial from a privacy standpoint as argued in [1].
>         To clarify, we used the test distribution to explain what unlearning would mean empirically. We have updated the statement to: "Empirically, these parameters must satisfy  $f(x_i, \theta^*) \simeq  f(x_i, \theta^*_{f})$ for $(x_i, y_i) \in \mathcal{D}_{\text{test}}$​." We agree with the reviewer that one could also define this as: "Mathematically, these parameters must satisfy $f(x, \theta^*) \simeq f(x, \theta^*_f)$ for $(x, y) \in \mathcal{D}$, where $\mathcal{D}$ is the true data distribution."
>
> 3. **About Table 1**
>     - **Variation in CIFAR10 VGG11_BN for kurmanji** - As per reviewer d7Sc's suggestions, we tuned the learning rate for SCRUB for CIFAR10 and CIFAR100 and have updated the results in the paper. This has reduced the standard deviation in the results of CIFAR10 for VGG11.
>     - **Architectural differences**- The difference between the two architectures could potentially be due to the different learning dynamics for the respective architectures. As we observed in the case of ViT's acceptable performance was obtained only when we increased the number of epochs which consequently led to a compute increase.
>     - **Bold fonting** - In our experiments, we tuned the hyperparameters to have high $ACC_{r}$ and low $ACC_{f}$ for all the algorithms (see equation 3). We believe the hyperparameters can be tuned to change the performance of an algorithm with respect to a particular metric in the table. Figure 4, which shows the effect of varying $\alpha$'s clarifies this further.
>
> 4. **About ViTs** - We have added the results for variations of ViT on ImageNet for our dataset in the table below. We add this to the appendix (See Table 11). We initially restricted the compute for Kurmanji et al (by restricting maximization steps to 2 and minimization steps to 3). We observed that increasing the compute budget (maximization steps to 5 and minimization steps to 10) leads to better retain and forget accuracies. We believe this is potentially due to an extremely low compute budget for an Iterative Gradient Descent/Ascent-based algorithm. A low number of steps forces the algorithm to have higher ascent steps to unlearn the forget-set, which is detrimental to the retain-set. Additionally, low maximization/minimization steps particularly affect ViTs as they have low inherent bias and require a larger number of iterations to converge.
>     - Table : ImageNet Class Unlearning for various ViT models.
>         |Method|Original Accuracy |  Acc_r      | Acc_f     |MIA     |
>         |------- | ------- | ------- | ---------- | ---------- |
>         | | | || |
>         | ViT_B_16 |$80.01$| $78.47\pm0.84$ | $0.2\pm0.63$ | $99.98\pm0.05$|
>         | ViT_L_16 |$78.83$| $78.48\pm.26$ | $0.22\pm0.67$ | $99.78\pm0.01$ |
>         | ViT_H_14 |$85.48$| $84.35\pm0.40$ | $0.2\pm0.63$ | $97.0\pm0.05$ |
>         | ViT_B_32 |$73.48$| $71.02\pm0.77$ | $0.4\pm0.84$ | $98.0\pm0.02$ |
>         | ViT_L_32 |$75.00$| $73.64\pm0.36$ | $0.2\pm0.63$ | $97.4\pm0.03$ |
>         | | | | | || | |

---

> ### Author Response · Authors · 2024-06-23
> **Continuation Author's Response to Reviewer gSKS's feedback - Part 2**
>
> 5. **Choice of alpha**- Below we present the set of hyperparameters that work best for the classes we studied.  We add this to Apenddix (See Table 7).
>     - Table : CIFAR Class Unlearning hyperparameter for both VGG11_BN and ResNet18.
>         |Dataset|  $\alpha_r$ |$\alpha_f$ |
>         |-------  | ----------   | --------- |
>         |        | |  |
>         |CIFAR10 | 100|  3|
>         |CIFAR100 | 1000|  30 |
>
>     - Table : ImageNet Class Unlearning.
>
>         |Method|  $\alpha_r$ |  $\alpha_f$|
>         |-------  | ----------   | --------- |
>         |VGG11_BN | 3000|  30|
>         |ViT_B_16 / ViT_L_16 | 100|  10 |
>         | ViT_H_16/ViT_B_32 / ViT_L_32 | 300|  30 |
>
>     Additionally, the importance scaling hyper-parameters could potentially be optimized using SGD. We could freeze all other network weights and minimize a combined cross-entropy loss (cross_entropy(retain) - cross_entropy(forget)) to optimize the alphas using gradient-based techniques.
> 6. **More saliency-based analysis figures like Figure 6** - We have added these in the appendix A10.
> 7. **Effect of having incomplete retain set** - We have added an experiment to study the effect of incomplete retain set below. In the table below we present ACC_r, ACC_f, the change in accuracy for the missing retain set and confusion of the missing class with the forget-class(cat class) obtained from Figure 8a. We observe that there is a maximum drop in accuracy when the dog class is missing from the retain set as it is maximally confused with the target cat class. Removal of cat class without access to the dog information catastrophically affects the model usability on dog class. We add this analysis in the main paper (see Table 4).
>
>     - Table : CIFAR10 Cat Class Unlearning with incomplete retain set on VGG11_BN. (hyper-parameter $\alpha_r=3000$, and $\alpha_f=30$)
>
>         |Missing Retain Set Class|  Acc_r      |  Acc_f     | ACC Change for Missing Class | % Confusion |
>         |-------  | ----------   | --------- |  --------- | --------- |
>         | Class 1 (airplane) |  93.88  | 0.0 | -0.1 | 0.8 |
>         | Class 2 (automobile) | 93.71 | 0.0 | +0.3 | 0.2|
>         | Class 3 (bird) | 93.72 | 0.0 | +0.2 | 3.1 |
>         | Class 5 (deer) | 93.74 | 0.0 | -0.4 | 1.9 |
>         | Class 6 (dog) |  92.36 | 0.0 | -13.7 | 8.0 |
>         | Class 7 (frog) |  93.77  | 0.0 | -1.0 | 2.0 |
>         | Class 8 (horse) | 93.82  | 0.0 | -0.2 | 0.8 |
>         | Class 9 (ship) |  93.79 | 0.0 | -1.7 | 0.3 |
>         | Class 10 (truck) |  93.91  | 0.0 | -0.6 | 0.6 |
>
> 8. **Inter-dependency of classes** - ImageNet dataset already captures the behavior the reviewer mentions. We evaluate our algorithm for unlearning class "Tibetan Terrier" ( class 200 ) and there are a total of 118 dog breeds in the ImageNet datasets (class 151	"Chihuahua" to class 268	"Mexican hairless").
>     Below we present the results for removing "Tibetan Terrier" from a pretrained ViT models. We get the top 5 most confusing classes from the confusion matrix on the test set of the model before unlearning and report the average accuracy of these classes before and after Unlearning "Tibetan Terrier". As the reviewer suspects, we observe a drop in the accuracy of the most confusing dog breeds to the "Tibetan Terrier" post unlearning. However, as the overall retain accuracy does not drop much indicating that unlearning increased the accuracy for some of the classes to compensate for the drop in accuracy for these confusing classes. We add these results in Table 3 and Table 14.
>
>     Additionally, this also explains the decrease in correctly classified dog class samples (only class to show decrease in accuracy) when we unlearn cat class from VGG11 model as shown in Figure 8.
>
>     - Table : ImageNet Class Unlearning for "Tibetan Terrier" (class 200)
>
>         |Model|Original Accuracy |  Acc_r      | Acc_f     |MIA     | Confusing dog breed Acc. Orignal| Confusing dog breed Acc. Unlearnt |
>         |------- | ------- |------- | ---------- | ---------- |---------- |---------- |
>         | ViT_B_16 | $80.01$| $78.92$ | $0$ |  $100$|80.8|54.4|
>         | ViT_L_16 | $78.83$| $78.61$ | $0$ | $100$ |79.6|68.0|
>         | ViT_H_14 |$85.48$ | $83.91$ | $0$ | $100$ |69.6|25.6|
>         | ViT_B_32 |$73.48$| $70.25$ | $0$ | $100$ |67.6|30.4|
>         | ViT_L_32 |$75.00$| $73.47$ | $0$ | $98.0$ |79.2|54.0|
>         | | | | | | | |
> ## Minor Issues
> - We have addressed several typos in our work, including the one mentioned by the reviewer.
>
> ## Reference
> [1] Inexact Unlearning Needs More Careful Evaluations to Avoid a False Sense of Privacy, ArXiv 2024

---

> > ### Comment · Reviewer_gSKS · 2024-07-07
> >
> > I thank the authors for their thorough rebuttal. My concerns have been alleviated, and I am recommending this paper for acceptance.

---

### Decision · Action_Editor_rhyy · 2024-07-17

**Recommendation:** Accept with minor revision

**Comment:**

The reviewers found the proposed method novel (Reviewer gSKS, Reviewer d7Sc), the paper well-written (Reviewer d7Sc) and the results show strong performance over class unlearning baselines in several settings ("works well on image classification tasks", Reviewer 6825), including a setting using a Vision Transformer where this method produces impressive results over baselines (Reviewer gSKS), scaling class-level unlearning to ImageNet and Vision Transformers (Reviewer d7Sc). The paper also presents nice ablations and analyses (Reviewer gSKS), makes "a commendable effort to analyze the effect of the method" (Reviewer d7Sc).

During the rebuttal, the authors have adjusted claims to address reviewer feedback on overclaimed novelty and overclaimed efficiency, and added several new experiments and analyses to address feedback (more thorough tuning of hyperparameters in some cases, investigating an incomplete retain set, inter-dependency of classes, more saliency-based analyses). They have also added to the discussion of limitations to address concerns raised by reviewers of trivial baselines for unlearning for some of the metrics used in this work.

Minor revisions:
1) "A strong MIA attack is proposed by Hayes et al. (2024), however, this attack requires substantial computational resources and are not practical for evaluating large datasets on large models hence we evalutated the unlearning on simple MIA." -- this comment seems outdated since the authors have added results with the attack from Hayes et al. now if I understand correctly. Please remove or adjust accordingly.

2) "In some of our experiments, we observe that the NegGrad+ approach outperforms the SoTA benchmarks" -- please relax the claim that "Our algorithm demonstrates SOTA performance in class unlearning setup with access to very few samples from the training dataset" (in the Introduction), to make it more specific by stating which settings SOTA is obtained (as it's not always the case).

3) Finally, please adjust the limitations section to explicitly mention the existence of the baseline suggested by Reviewer 6825 and Reviewer d7Sc, and the associated discussion.

**Audience:**

This paper proposes an unlearning method which is of interest to the TMLR community.

Reviewer 6825 and Reviewer d7Sc both pointed out a simple baseline that would work perfectly on the evaluation setup of the paper: simply set the output weight corresponding to the class(es) to unlearn to zero (or a large number, or ignore that softmax position when decoding). As the authors pointed out, that trivial baseline may satisfy some metrics (e.g. accuracy-based metrics) but not others (e.g some types of attacks), since information about the forget set classes can still live in the trained weights. The existence of this trivial baseline then does not invalidate methods like the one proposed here but it raises doubts about the evaluation metrics used and problem formulations. Overall, I believe that, having acknowledged this limitation of evaluation procedures in the paper, the proposed method is still of interest to the community.

**Claims And Evidence:**

This paper proposes a class unlearning method inspired by continual learning methodology. The goal is to unlearn a class (or multiple classes) using a few samples from those classes, where the goal of unlearning is defined (informally) as producing parameters that are indistinguishable from those that would have obtained if training from scratch without examples of the "forget class(es)". Empirically, they assess success by comparing the accuracy of the unlearned and retrained models on retain and forget samples. They also conduct Membership Inference Attacks, including a strong one recently proposed for unlearning.

Their approach estimates the "activation space" for the retain set and the forget set (by computing SVD on the activations of retain and forget samples, respectively, for each layer separately), using a small subset of retain and forget examples. Using these spaces, they isolate the class-discriminatory space (computed by removing from the forget space the information that is shared between the retain and forget sets). They then obtain the unlearned model from the original model by modifying the activations of each layer to suppress class-discriminatory information (by post-multiplying them with an appropriately-constructed matrix). This approach to unlearning is training-free and does not require gradient updates.

The authors' claims that their method "exhibits competitive unlearning performance and resilience against Membership Inference Attacks (MIA)" are well-substantiated with empirical evidence, as the reviewers pointed out. During the rebuttal, the authors have amended claims that were unsubstantiated or incorrect (e.g. acknowledging other training-free unlearning algorithms from the literature, providing empirical evidence for efficiency claims). The concerns that the reviewers raised on this front have been mostly addressed (see comments section).

---

> ### Author Response · Authors · 2024-07-19
> **Requested Minor Changes**
>
> Below we address the requested changes -
> 1. We removed the line - "A strong MIA attack is proposed by Hayes et al. (2024), however, this attack requires substantial computational resources and are not practical for evaluating large datasets on large models hence we evalutated the unlearning on simple MIA."
> 2. We clarify the claim in the introduction - "In a class-unlearning scenario, our algorithm competes with or outperforms several algorithms studied in this work, despite having access to very few samples from the training dataset (less than $4\%$ for all our experiments)."
> 3. We update the limitation as suggested - "Further, machine unlearning is an evolving field, and current evaluation techniques can be misled by naive baselines, such as setting the bias term to a large negative number or hard-coding the network to output nothing (or a random class) if the prediction is the forget class. These trivial baselines might satisfy certain metrics (e.g., accuracy-based metrics) but fail others (e.g., specific types of attacks), as information about the forget set classes can still persist in the trained weights. This suggests that existing evaluations for unlearning may be insufficient to guarantee privacy . We hope future works will address these shortcomings in evaluating the privacy provided by unlearning algorithms."